# On Uncertainty Calibration for Equivariant Functions

**Edward Berman** *                                                              *berman.ed@northeastern.edu*
*Department of Mathematics, Northeastern University*

**Jacob Ginesin**                                                                *ginesin@cmu.edu*
*Carnegie Mellon University*

**Marco Pacini**                                                                 *mpacini@fbk.eu*
*University of Trento & Fondazione Bruno Kessler*

**Robin Walters**                                                                *r.walters@northeastern.edu*
*Khoury College of Computer Sciences, Northeastern University*

**Reviewed on OpenReview:** *https://openreview.net/forum?id=rxLUTPLBT3*

## Abstract

Data-sparse settings such as robotic manipulation, molecular physics, and galaxy morphology classification are some of the hardest domains for deep learning. For these problems, equivariant networks can help improve modeling across undersampled parts of the input space, and uncertainty estimation can guard against overconfidence. However, until now, the relationships between equivariance and model confidence, and more generally equivariance and model calibration, have yet to be studied. Since traditional classification and regression error terms show up in the definitions of calibration error, it is natural to suspect that previous work can be used to help understand the relationship between equivariance and calibration error. In this work, we present a theory relating equivariance to uncertainty estimation. By proving lower and upper bounds on uncertainty calibration errors (ECE and ENCE) under various equivariance conditions, we elucidate the generalization limits of equivariant models and illustrate how symmetry mismatch can result in miscalibration in both classification and regression settings. We complement our theoretical framework with numerical experiments that clarify the relationship between equivariance and uncertainty using a variety of real and simulated datasets, and we comment on trends with symmetry mismatch, group size, and aleatoric and epistemic uncertainties.

## 1 Introduction

Equivariant neural networks are a class of neural networks that encode group symmetries into the structure of the network architecture so that the symmetries do not need to be learned from data. Understanding both model calibration and confidence is particularly useful in the data-sparse settings where equivariant neural networks tend to thrive, such as pick-and-place robotics tasks (Kalashnikov et al., 2018; Wang et al., 2022b;a; Fu et al., 2023; Huang et al., 2023; 2024b;a; Wang, 2025), galaxy morphology classification (Pandya et al., 2023; 2025a), and molecular physics (Zou et al., 2023; Ramakrishnan et al., 2014). While equivariance has proved effective in these domains, it does have some drawbacks, including diminishing benefits at scale as in the case of extremely large datasets or aggressive augmentation (Wang et al., 2023b; Klee et al., 2023; Gruver et al., 2023; Brehmer et al., 2024; Abramson et al., 2024), provable degradation on model performance in cases of symmetry mismatch (Wang et al., 2024), more complex architectures, and higher compute costs.

Despite these drawbacks, a surprising result of Wang et al. (2023a) is that equivariant neural networks can still be effective even in cases of mismatch between the model and the data symmetry. This finding

---

*Work from all authors done with the Geometric Learning Lab

motivated the work of Wang et al. (2024), which explored how equivariance can affect model *accuracy*, both positively and negatively. However, it is not yet understood how equivariance impacts model *calibration*, loosely defined as the disagreement between a model's accuracy and predicted *confidence.*

To help understand the tradeoffs associated with equivariance, we seek to quantify the impact of equivariance on different metrics, namely expected calibration error for classification and expected normalized calibration error for regression. The only works that directly examine the link between equivariance and calibration are Sun et al. (2023) and Cherif et al. (2024), although these are purely experimental and provide only a few illustrative cases where equivariance influences model calibration. Despite this, because different notions of calibration error involve expressions corresponding to classification or regression errors themselves, previous results on the generalization limits of equivariant models can be applied to the study of calibration error. Understanding this relationship allows us to address several unanswered questions on the subject of calibration and confidence of equivariant models. Namely, when does equivariance help a model predict its own confidence? How do notions of symmetry mismatch between model and data affect model calibration? What is the general relationship between equivariance and uncertainty estimation? Beyond raw performance, answering these questions is crucial for developing reliable symmetry-aware models in safety-critical and high-stakes applications.

The purpose of this work is to address both the lack of a theory relating equivariance to uncertainty estimation, and the scarcity of experiments exploring this relationship in practice. To accomplish this, we extend the error bounds given by Wang et al. (2024) to a broader class of calibration losses. In this way, we can quantify the effect of equivariance not just on accuracy, but also on calibration. In particular, we show that calibration error is related to typical classification and regression errors over the preimage of each confidence prediction. These errors have known bounds for equivariant functions, which we use to provide lower and upper bounds on various calibration error metrics. We explore these threads in the context of the equivariance taxonomy from (Wang et al., 2024), which describes scenarios when the ground truth function has the same symmetry as the equivariant model, the ground truth function disagrees with the symmetry in the model, and the model symmetry transforms in-distribution data to out-of-distribution data. We also define a new metric, *the aleatoric bleed*, that prescribes how well a model is able to distinguish between different types of uncertainty. We provide a lower bound on aleatoric bleed for equivariant models. Our study illustrates that the effect of equivariance on model calibration is dependent on the extent of the symmetry mismatch, a finding similarly reported for accuracy in Wang et al. (2023a; 2024). We also perform experiments on a wide variety of real and simulated datasets, empirically showing that symmetry mismatch can increase calibration error and demonstrating that our theoretical findings are informative in practical settings.

We summarize our contributions as follows:

1. For *classification*, we provide bounds on calibration error and we tighten the bound in the limiting case of an invariance (Section 4).

2. For *regression*, we generalize expected normalized calibration error beyond scalar values for mean and variance predictions. We derive its upper bound on certain equivariant models (Section 5) and derive its lower bound in the special case of scalar-valued mean and variance predictions.

3. Additionally, we define a new metric, the *aleatoric bleed*, which quantifies miscalibration in terms of aleatoric and epistemic uncertainty (Section 6).

4. We provide *illustrative examples* and run *numerical experiments* on diverse real and simulated datasets. We find model calibration and aleatoric bleed suffer in cases of symmetry mismatch, and show our bounds provide a useful way of assessing relative model calibration errors.

## 2 Related Work

**Equivariant Learning.** Our work studies the problem of symmetry mismatch between model and data through the lens of approximation error. We are closest thematically to Petrache & Trivedi (2023) and Wang et al. (2024), which both establish bounds on function generalization under various assumptions of symmetry

mismatch, but neither study calibration errors specifically. We also consider the equivariance taxonomy established in Wang et al. (2024), but depict its ramifications on uncertainty quantification. Throughout this paper, we assume access to universal $G$-equivariant models, guaranteed by prior work (Maron et al., 2019; Yarotsky, 2022). Yet, universality alone does not ensure reliable uncertainty estimates, and therefore our focus is on deriving error bounds for calibration error. In order to prove these bounds, our work also uses the strategy of decomposing the input and output spaces, which we accomplish through taking the quotient by the symmetry group. This is a strategy similarly employed in Sannai et al. (2021), Lawrence (2022), Petrache & Trivedi (2023), and Wang et al. (2024). Theory on the relevant group representations for functions that output probabilities is partly addressed in Bloem-Reddy et al. (2020) and Dobriban & Yu (2025), but no bounds on expected calibration error are presented. Reasoning about distributions in terms of invariants also has a rich history in deriving uninformative priors for Bayesian analysis – see Jaynes (1968). We build on this by studying how equivariance can affect the reliability of Bayesian methods. Specifically, we look at the ability to separate different types of uncertainty using evidential regression, including on multivariate distributions, generalizing some of the work of Van der Linden et al. (2025) who study equivariant model selection when trained to predict a univariate distribution. We also build off of Gelberg et al. (2024) by studying the behavior of Bayesian models under various equivariance constraints other than the weight space permutation invariances they consider.

**Aleatoric and Epistemic Uncertainties.** A longstanding goal in the computational sciences is to separate model (epistemic) uncertainties from (aleatoric) uncertainties inherent to the data (Ulmer et al., 2021; Hüllermeier & Waegeman, 2021; Osband et al., 2023; Fuchsgruber et al., 2024; Chau et al., 2025). Previous work underscores how this separation can be difficult to perform in practice. Techniques such as those presented in Kendall & Gal (2017); Amini et al. (2020) often fail to distinguish these uncertainties due to effects such as loss shaping (Ovadia et al., 2019; Valdenegro-Toro & Mori, 2022; Osband et al., 2023; Wimmer et al., 2023; Nevin et al., 2024; Jürgens et al., 2024). Our work explores this problem in the context of symmetry. In particular, we explore how the epistemic uncertainty—the uncertainty often quantified by calibration errors—can be confused with aleatoric uncertainty due to symmetry mismatch.

**Learning Parameterized Distributions.** For learning tasks with inherent uncertainty, it is natural to design a neural network that approximates a probability distribution rather than a single point estimation. There are many ways of doing this, such as with Bayesian Neural Networks (Kononenko, 1989), Epistemic Neural Networks (Osband et al., 2023), Normalizing Flows (Rezende & Mohamed, 2015; Kobyzev et al., 2020; Papamakarios et al., 2021), or using the softmax function (Goodfellow et al., 2016) to learn a categorical distribution. These approaches are often computationally expensive to train and sample in practice. Thus, we often employ parameterization techniques to constrain neural networks to output simplified probability distributions and train them using a negative log likelihood loss derived from said distribution. For example, mean variance estimators (MVE) are the simplest type of neural network that predicts a parameterized distribution. Instead of predicting a single output, MVEs predict a mean $\mu$ and a variance $\sigma^2$ (Nix & Weigend, 1994; Seitzer et al., 2022). There is also work extending MVEs to learn covariances for multivariate distributions (Tomczak et al., 2020) and linear combinations of Gaussians (Diakonikolas et al., 2020). Amini et al. (2020) extend MVEs by imposing a prior on $\mu$ and $\sigma^2$ and performing evidential regression, which in-turn provides enough parameters to disentangle aleatoric and epistemic uncertainties. A unique feature of our work is that we use models that predict parameterized distributions in order to define calibration error in a regression setting and also to experimentally test the effect of equivariance on the ability to learn a reliable uncertainty estimate.

**Calibration Error.** In classification, probabilistic models (e.g. logistic regression or softmax classifiers) output a distribution over labels, and the predicted label is chosen as the one with maximum probability. However, even if a model gives a label $y$ the highest probability $p$, that does not mean the model will necessarily be correct with probability $p$. This mismatch is often quantified with the expected calibration error (ECE) (Guo et al., 2017) and is often estimated using binning procedures that approximate the continuous push-forward density of different confidence regions. Miscalibration can analogously be measured for regression tasks (Pernot, 2023; Levi et al., 2022b). The idea is to compare true labels $y$ with a predicted mean $\mu$ and variance $\sigma^2$ of a MVE. One should expect the squared errors $(y - \mu)^2$ to average out to the variance

$\sigma^2$. This idea was made precise by Levi et al. (2022a), who proposed the expected normalized calibration error (ENCE) to quantify this exact discrepancy. A key limitation of their work is that it is formulated in terms of binning approximations rather than in terms of a continuous probability density. Another key limitation of their work is that they assume mean and variance are scalar values. We generalize these metrics for continuous densities and multivariate normal distributions. Beyond ECE and ENCE, some works propose calibration objectives and training procedures in terms of coverage (Gneiting & Raftery, 2007; Lemos et al., 2023; Sun et al., 2023), distributional calibration (Kuleshov et al., 2018), and post-hoc variance scaling (Laves et al., 2020). We focus on ECE and ENCE because they are clearly formulated objectives that can be studied independently of the training process itself. The work of Sun et al. (2023) suggests that equivariance can improve model calibration, but a theoretical justification for this is not present in the literature and is something we comment on in this work.

## 3 Background

We review the definition of equivariance and how symmetry constraints of a model class may be mismatched with a given dataset. Additionally, we review evidential regression, a technique for learning model and data uncertainties as distinct outputs of a neural network.

### 3.1 Equivariance

Here, we give precise definitions of equivariance and invariance. For a general review of the mathematical background, we direct the reader to Artin (1998); Hall (2013); Esteves (2020).

Let $\phi : X \to Y$ be a map between input and output vector spaces $X$ and $Y$. Let $G$ be a group with representations $\rho^X$ and $\rho^Y$ which transform vectors in $X$ and $Y$ respectively. Representations are group homomorphisms which map group elements to invertible linear transformations. When clear, we omit the representation map and write $gx$ for $\rho^X(g)x$. The map $\phi : X \to Y$ is *equivariant* if

$$\rho^Y(g)\phi(x) = \phi(\rho^X(g)x) , \text{ for all } g \in G, x \in X .$$

Invariance is a special case of equivariance in which $\rho^Y = \text{Id}^Y$ for all $g \in G$. I.e., *a map $\phi : X \to Y$ is invariant if it satisfies*

$$\phi(x) = \phi(\rho^X(g)[x]) , \text{ for all } g \in G, x \in X.$$

Thus, with an invariant operator, the output of $\phi$ is unaffected by transformations applied to the input.

**Fundamental Domain.** This paper will use iterated integration over an orbit and a set of orbit representatives, which we call the fundamental domain.

**Definition 1** (**Definition 4.1 in Wang et al. (2024)**)**.** Let $d$ be the dimension of a generic orbit of $G$ in $X$ and $n$ the dimension of $X$. Let $\nu$ be the $(n - d)$ dimensional Hausdorff measure in $X$. A closed subset $F$ of $X$ is called a fundamental domain of $G$ in $X$ if $X$ is the union of conjugates of $F$, i.e., $X = \cup_{g \in G} gF$, and the intersection of any two conjugates has 0 measure under $\nu$.

In the following, we give a simple example of fundamental domain.

**Example 1.** Let $G = \text{SO}(n)$, $X = \mathbb{R}^n$ and $F = \{(x, 0, \ldots, 0) \mid x \in \mathbb{R}_{\geq 0}\}$. The closed set $F$ is a fundamental domain of $\text{SO}(n)$ in $\mathbb{R}^n$. Indeed, the intersection $g_1 F \cap g_2 F = \{0\}$ has measure 0 for each distinct $g_1, g_2 \in G$ and $\mathbb{R}^n = \cup_{g \in \text{SO}(n)} gF$.

### 3.2 Equivariant Learning

Following Wang et al. (2024), we first establish an equivariant learning setting that describes assumptions on the ground truth and hypothesis class. We work in the deterministic realizable case of statistical learning theory: data distribution is defined on $X$ and given by the probability density function $p \colon X \to \mathbb{R}$, labels are given deterministically by a ground truth function $f \colon X \to Y$. Following this standard statistical

learning framework, we assume that training and testing samples are drawn i.i.d. from the same underlying distribution $p$, i.e., no distribution shift occurs at test time. The goal for a function space $\mathscr{H} = \{h \colon X \to Y\}$ is to fit the function $f$ by minimizing an error function $\mathrm{err}(h)$. Let $\mathbb{1}(x)$ be an indicator function that equals 1 if the condition in the argument is satisfied and 0 otherwise. In classification, $\mathrm{err}(h)$ is the classification error rate; for regression tasks, the error function is a $L_2$ norm function,

$$
\begin{aligned}
\mathrm{err}_{\mathrm{cls}}(h) &= \mathbb{E}_{x \sim p}\left[\mathbb{1}(f(x) \neq h(x))\right], \\
\mathrm{err}_{\mathrm{reg}}(h) &= \mathbb{E}_{x \sim p}\left[\left\|h(x) - f(x)\right\|_2^2\right].
\end{aligned}
$$

### 3.3 A Taxonomy of Equivariance: Correct, Incorrect, and Extrinsic.

Wang et al. (2024) establish a taxonomy which describes the relationship of the symmetry of the function space to the symmetry in the data. We review their definitions of correct, incorrect, and extrinsic equivariance. These definitions help us understand the ability of equivariant functions to approximate datasets that may or may not have the same symmetries. An important part of this taxonomy is extrinsic symmetry, which describes the case where the action of the group moves data points out of the support of their original distribution.

**Definition 2** (**Point-wise Correct, Incorrect, and Extrinsic Equivariance, Definitions 3.5-3.7 in Wang et al. (2024)**). Assume $h$ is equivariant with respect to a group $G$. For $g \in G$ and $x \in X$ where $p(x) \neq 0$, if $p(gx) \neq 0$ and $f(gx) = gf(x)$, $h$ has *correct equivariance* with respect to $f$ at $x$ under transformation $g$. For $g \in G$ and $x \in X$ where $p(x) \neq 0$, if $p(gx) \neq 0$ and $f(gx) \neq gf(x)$, $h$ has *incorrect equivariance* with respect to $f$ at $x$ under transformation $g$. For $g \in G$ and $x \in X$ where $p(x) \neq 0$, if $p(gx) = 0$, $h$ has *extrinsic equivariance* with respect to $f$ at $x$ under transformation $g$.

If $f$ has point-wise correct equivariance for all $x \in X$, $g \in G$, then we say $f$ has correct equivariance. The same follows for incorrect and extrinsic equivariance as well. In the case that $f$ has correct equivariance, we assume $f$ lies in the considered hypothesis class $\mathscr{H}$. This assumption aligns with Wang et al. (2024) and is realistic, since several universality results for equivariant models are already known (Yarotsky, 2022).

### 3.4 Error Bounds for Equivariant Models in Classification and Regression Tasks

Our goal is to generalize the bounds from Wang et al. (2024) to a calibration objective. We review the main results from Wang et al. (2024) here. Given that equivariance is not always correct, the following definitions and theorems detail how symmetry mismatch can harm model fitting for classification or regression problems. We start with invariant classification.

**Definition 3** (Majority Label Total Dissent). For the orbit $Gx$ of $x \in X$, the total dissent $k(Gx)$ is the integrated probability density of the elements in the orbit $Gx$ having a different label than the majority label:

$$
k(Gx) = \min_{y \in Y} \int_{Gx} p(z)\mathbb{1}(f(z) \neq y)dz.
$$

**Theorem 1** (**Theorem 4.3 in Wang et al. (2024)**). The error $\mathrm{err}_{\mathrm{cls}}(h)$ is bounded below by $\int_F k(Gx)dx$.

We now detail the relevant error lower bound assuming invariance in the regression setting.

**Theorem 2** (**Theorem 4.8 in Wang et al. (2024)**). Assume $h$ is $G$ invariant so that $h(gx) = h(x)$ for all $g \in G$. Assume $Y = \mathbb{R}^n$. Denote by $p(Gx) = \int_{z \in Gx} p(z)dz$ the probability of the orbit $Gx$. Denote by $q(z) = \frac{p(z)}{p(Gx)}$ the normalized probability density of the orbit $Gx$ such that $\int_{Gx} q(z)dz = 1$. Let $\mathbb{E}_{Gx}[f]$ be the mean of the function $f$ on the orbit $Gx$ defined, and let $\mathbb{V}_{Gx}[f]$ be the variance of $f$ on the orbit $Gx$,

$$
\begin{aligned}
\mathbb{E}_{Gx}[f] &= \int_{Gx} q(z)f(z)dz = \frac{\int_{Gx} p(z)f(z)dz}{\int_{Gx} p(z)dz}, \\
\mathbb{V}_{Gx}[f] &= \int_{Gx} q(z)\|\mathbb{E}_{Gx}[f] - f(z)\|_2^2 dz.
\end{aligned}
$$

We have $\mathrm{err}_{\mathrm{reg}}(h) \geq \int_F p(Gx)\mathbb{V}_{Gx}[f]$.

Theorem 3 generalizes this to the setting of equivariance. Before stating the theorem, we establish how to lift an integral over an orbit to an integral over the group. Details on necessary hypotheses for this to be well defined are given in Appendix A. Denote the stabilizer by $G_x = \{g : gx = x\}$. Denote by $a_x : G/G_x \to Gx$ the identification of the orbit $Gx$ and coset space $G/G_x$ with respect to the stabilizer. We have

$$\int_{Gx} f(z)dz = \int_G f(gx)\alpha(g,x)dg$$

where

$$\alpha(g,x) = \left(\int_{Gx} dh\right)^{-1} \left| \frac{\partial a_x(\bar{g})}{\partial \bar{g}} \right|.$$

We may now state the equivariant regression lower bound:

**Theorem 3** (**Theorem 4.9 in Wang et al. (2024)**). Assume $h$ is equivariant: that is, $h(\rho_X(g)x) = \rho_Y(g)h(x)$ where $g \in G$, $\rho_X$ and $\rho_Y$ are group representations. Denote $\rho_X(g)x$ and $\rho_Y(g)y$ by $gx$ and $gy$. Let $Y = \mathbb{R}^n$ and Id be the identity. Define the matrix $Q_{Gx} \in \mathbb{R}^{n\times n}$ and $q(gx) \in \mathbb{R}^{n\times n}$ so that $\int_G q(gx)dg = $ Id by

$$
\begin{aligned}
Q_{Gx} &= \int_G p(gx)\rho_Y(g)^T\rho_Y(g)\alpha(x,g)dg, \\
q(gx) &= Q_{Gx}^{-1}p(qx)\rho_Y(g)^T\rho_Y(g)\alpha(x,g).
\end{aligned}
$$

If $f$ is equivariant, $g^{-1}f(gx)$ is a constant for all $g \in G$. Define

$$\mathcal{E}_G[f,x] = \int_G q(gx)g^{-1}f(gx)dg.$$

The error of $h$ has lower bound $\mathrm{err}_{\mathrm{reg}}(h) \geq \int_F \int_G p(gx)\|f(gx) - g\mathcal{E}_G[f,x]\|_2^2\alpha(x,g)dgdx$.

### 3.5 Evidential Regression

This work studies the effect of equivariance on the ability to separate model and data-centric uncertainties. We now describe evidential regression, a learning framework that allocates mass between mean and dispersion under a surrogate loss. That is, a loss function determines how much uncertainty should be allocated to the model and how much should be allocated to the true dispersion (e.g. variance) of the data. The allocation is not identifiable and is sensitive to loss shaping and misspecification. We review the relevant theory and notation from section 2.3 in Hüllermeier & Waegeman (2021) for defining different sources of uncertainty, then Amini et al. (2020) for describing evidential regression.

The following definitions make the notions of model and data-centric uncertainties concrete:

**Aleatoric Uncertainty:** Aleatoric uncertainty refers to the irreducible part of the uncertainty. Given spaces $X$ and $Y$ and an instance $x_q \in X$, the aleatoric uncertainty is the spread in $p(y|x_q)$.

**Epistemic Uncertainty:** Model uncertainty and approximation uncertainty, on the other hand, are subsumed under the notion of epistemic uncertainty. Let the spaces $X$ and $Y$ be the same as before. Let $l : Y \times Y \to \mathbb{R}$ be the loss function and let $f^*$ be the associated point-wise Bayes predictor defined as

$$f^*(x) := \arg\min_{\hat{y}\in Y} \int_Y l(y, \hat{y})dP(y|x). \tag{1}$$

Epistemic uncertainty is the uncertainty due to the lack of knowledge of the perfect predictor Equation 14.

**Evidential Regression.** We now describe evidential regression, which prescribes a specific parameterization of the two uncertainties. Given $(y_1, \ldots, y_n) \sim \mathcal{N}(\mu, \sigma^2)$, we may impose priors

$$
\begin{aligned}
\mu &\sim \mathcal{N}(\gamma, \sigma^2\nu^{-1}) \\
\sigma^2 &\sim \Gamma^{-1}(\alpha, \beta)
\end{aligned}
$$

where $\Gamma(\cdot)$ is the gamma function. Let $m = (\gamma, \nu, \alpha, \beta)$, and $\gamma \in \mathbb{R}$, $\nu > 0$, $\alpha > 1$, $\beta > 0$. One can then show that $p(y_i|m) = St(y_i; \gamma, \frac{\beta(1+\nu)}{\alpha\nu}, 2\alpha)$, where the St distribution has probability density given by

$$St(t; \mu, \sigma, \nu) = \frac{\Gamma\left(\frac{\nu+1}{2}\right)}{\sqrt{\pi\nu}\sigma\Gamma\left(\frac{\nu}{2}\right)}\left(1 + \frac{1}{\nu}\left(\frac{t-\mu}{\sigma}\right)^2\right)^{-(\nu+1)/2}.$$

Parameterizing the Student's t distribution as a four parameter family is useful because it allows us to define our prediction, aleatoric uncertainty, and epistemic uncertainty in a rigorous way:

$$\begin{aligned}
\mathbb{E}[\mu] &= \gamma \quad \text{(Prediction)} \\
\mathbb{E}[\sigma^2] &= \frac{\beta}{\alpha-1} \quad \text{(Aleatoric Uncertainty)} \\
\text{Var}[\mu] &= \frac{\beta}{\nu(\alpha-1)} \quad \text{(Epistemic Uncertainty)}.
\end{aligned}$$

The prediction definition is clear in that it represents the expected mean of the Student's t distribution used to fit the data. Aleatoric uncertainty represents the uncertainty in $P(y|x)$, whereas the epistemic uncertainty represents uncertainty in the predictive law (the model). Practically, aleatoric uncertainty can be understood as the irreducible uncertainty due to inherent stochasticity in the data. In contrast, epistemic uncertainty is reducible uncertainty due to finite data, features, or misspecification. It vanishes in the limit of perfect knowledge and correct model class.

With some slight abuse of notation, we may abbreviate these as $\sigma^2_{\text{aleatoric}} = \mathbb{E}[\sigma^2]$ and $\sigma^2_{\text{epistemic}} = \text{Var}[\mu]$. Some works take these uncertainties to be additive. That is, the model will predict $\gamma \pm \sigma^2_{\text{aleatoric}} \pm \sigma^2_{\text{epistemic}}$. See Freedman et al. (2025) and Berman et al. (2025) for examples where this is done in different scientific domains. We do not assume these uncertainties to be additive in this work.

We note that there is an inherent identifiability issue: there are multiple such $m = (\gamma, \nu, \alpha, \beta)$ that can fit similar likelihoods with different aleatoric and epistemic allocations. This issue motivates our experiments on aleatoric bleed in Section 6.

## 4 Invariant Classification Calibration

In this section, we present our results on bounds of uncertainty calibration error for classification on invariant functions. In particular, we will show how miscalibration can be understood by examining the fibers of each uncertainty estimate and then studying the error on each of those fibers. The error on these fibers is related to the symmetries of the model and the data, which establishes a relationship between symmetry and model calibration. In the main, our bounds demonstrate how badly a function can become miscalibrated due to incorrect and extrinsic invariance. In particular, the lower and upper bounds are tightened by the dissent on individual fibers. Additionally, this section provides experimental results that show how incorrect invariance can affect model calibration in domains where the theory is less tractable. For practitioners who are more interested in empirical results and conclusions, experiments begin in Section 4.7.

### 4.1 Classification Problem

Consider a function $f : X \to Y$ where $Y$ is a finite set of labels. Let $q : X \to \mathbb{R}$ be a probability density on the domain $X$. We define a function space $\mathscr{H} = \{h : X \to Y \times [0, 1]\}$. If $h(x) = (h_Y, h_P)$ then $h_P$ represents the confidence estimate associated with the predicted label $h_Y$. The goal is to find the function $h \in \mathscr{H}$ that fits the function $f$ *and* to properly predict its own confidence by minimizing the expected calibration error (Equation 2, and Equation 2 in Guo et al. (2017)). Following Wang et al. (2024), we assume that the class $\mathscr{H}$ is arbitrarily expressive except that it is constrained to be equivariant with respect to a group $G$. In the classification setting, we specifically assume $h$ to be $G$-invariant. While not all classification problems are $G$-invariant, this is the case most commonly considered in the literature. Let $r(p)$ be the probability density such that $\mathbb{P}(p_1 \leq h_P(x) \leq p_2) = \int_{p_1}^{p_2} r(p)dp$. Equivalently, $r$ is the push-forward of $q$ over $h_P$. The expected calibration error is nominally defined

$$\text{ECE}(h) = \mathbb{E}_{h_P}\left[\left|\,\mathbb{P}\left(f = h_Y \mid h_P = p\right) - p\,\right|\right] \tag{2}$$

as in Guo et al. (2017). Intuitively, if a model has confidence $p$, then it should be accurate with probability $p$. This metric penalizes the discrepancy between accuracy and confidence averaged over all of confidences weighted by the push-forward density $r$. The definition in Guo et al. (2017) abuses notation slightly, in that the probability of any event drawn from a continuous random variable has probability zero, i.e., $p(h_P = p) = 0$ for all $p$. We can rectify this by defining ECE as

$$\text{ECE}(h) = \lim_{\varepsilon \to 0} \mathbb{E}_{p \sim r(p)}\left[\left|\,\mathbb{P}\left(f = h_Y \mid p - \varepsilon < h_P < p + \varepsilon\right) - p\,\right|\right], \tag{3}$$

however, we drop the limits for brevity throughout.

We abbreviate $\mathbb{P}(f(x) = h_Y(x)|h_P(x) = p)$ as $\text{Acc}_p(h)$.[1] This is the true accuracy of the model when the predicted confidence is $p$. Hence $h$ is well calibrated at confidence $p$ when $\text{Acc}_p(h) = p$, under confident when $\text{Acc}_p(h) > p$ and overconfident when $\text{Acc}_p(h) < p$. For the purposes of approximation, $|2\varepsilon|$ can be viewed as the bin width.

We briefly comment on the well-definedness of Equation 3. First, $\mathbb{P}\left(f = h_Y | p - \varepsilon < h_P < p + \varepsilon\right)$ is well defined when $r(p) \neq 0$ for all $p \in [0, 1]$. Moreover, we note that in general, if $\mathcal{C} = \{C = c\}$, it is not always permissible to define $\mathbb{P}(A|\mathcal{C}) = \lim_{\varepsilon \to 0}\mathbb{P}(A|c - \varepsilon < C < c + \varepsilon)$. This is because we face contradictions when $\mathcal{C} = \{C = c\} = \{D = d\}$, but the random variables $C$ and $D$ have different *densities* defined with respect to different *measures*. This results in contradictions where $\mathbb{P}(A|\mathcal{C}) = \lim_{\varepsilon \to 0}P(A|c - \varepsilon < C < c + \varepsilon)$ and $\mathbb{P}(A|\mathcal{C}) = \lim_{\varepsilon \to 0}\mathbb{P}(A|d - \varepsilon < D < d + \varepsilon)$ but $\lim_{\varepsilon \to 0}\mathbb{P}(A|c - \varepsilon < C < c + \varepsilon) \neq \lim_{\varepsilon \to 0}\mathbb{P}(A|d - \varepsilon < D < d + \varepsilon)$, see for example the Borel-Kolmogorov Paradox (Kolmogorov, 1933; Jaynes, 2003)[2]. In other words, the probability density conditioned on an event with zero probability can only be specified with respect to a given reference measure that determines the probability density function being conditioned on. Therefore, we must specify a measure on $X$ so that the *random variable $h_p$ has push-forward density $r(p)$* defined with respect to the *push-forward measure*. We express this with the following assumption.

**Assumption 1** (Hausdorff Measurability of input domain)**.** The input domain $X$ is equipped with an $|X|$ dimensional Hausdorff measure $\mathcal{H}$. The density $r(p)$ is the push-forward of $q(x)$ over $h_P$, meaning it is defined with respect to the accompanying push-forward measure $h_P \# \mathcal{H}$ on $[0, 1]$.

Assumption 1 is sufficient for Equation 3 to be uniquely defined. Our construction is supported by various disintegration theorems in the literature (e.g. Pachl, 1978; Chang & Pollard, 1997). For further background, see also Rokhlin (1949); Bogachev (2007). We also note that we do not need these well-definedness properties to hold in the special case where $r(p)$ is discrete or when we are computing approximations that treat $h_P$ as discrete. In each case, we average over the confidences (or confidence bins) with non-zero probability.

Since we assumed that there are finitely many labels in the co-domain $Y$, we can assume that $\mathbb{P}(h_Y(x)|h_P(x) = p)$ is a discrete probability distribution for each value of $p$. Therefore, each outcome in the distribution has a probability less than one.

## 4.2 ECE Upper Bounds

We now show that ECE is a bounded. Since ECE is the average of a random variable bounded between 0 and 1, ECE is also bounded between 0 and 1. However, we can improve upon this both with and without the assumption of invariance. Our first proposition concerns an unconstrained model, i.e., it makes no use of the assumption of invariance on $h$, however, the propositions that follow show how invariance can be used to tighten the lower and upper bounds. We will also consider special cases of binary classification.

To start, we consider an unconstrained model.

---

[1]Some works refer to $\text{Acc}_p(h)$ as the calibration function, e.g. Vaicenavicius et al. (2019)
[2]This paradox is most easily exemplified with the Great Circle Puzzle.

**Proposition 1.** ECE is bounded from above by $\frac{1}{2} + \int_0^1 r(p)|\frac{1}{2} - p|dp$.

*Proof.* See that $|\text{Acc}_p(h) - p| = |\text{Acc}_p(h) - p + \frac{1}{2} - \frac{1}{2}| \leq |\text{Acc}_p(h) - \frac{1}{2}| + |\frac{1}{2} - p| \leq |\frac{1}{2} - p| + \frac{1}{2}$. Therefore, $\int_0^1 r(p)(|\text{Acc}_p(h) - p|dp \leq \int_0^1 r(p)\left(|\frac{1}{2} - p| + \frac{1}{2}\right)dp = \frac{1}{2} + \int_0^1 r(p)|\frac{1}{2} - p|dp$. $\qquad\square$

The upper bound presented in the Proposition 1 is loose without any further assumptions. We now show that assuming the function space is invariant allows us to tighten the upper bound. The proof uses iterated integration over the orbits and the set of orbit representatives (the fundamental domain). Let us first make some assumptions in order for iterated integration on subsets of $X$ to be well defined.

**Assumption 2** (Smoothness and Separability Hypothesis)**.** For a group $G$ acting on a domain $X$, the union of all pairwise intersections $\cup_{g_1 \neq g_2}(g_1 F \cap g_2 F)$ have measure 0 and that $F$ and $Gx$ are differentiable manifolds for all $x \in X$. This holds for fibers $\mathcal{F}_p = h^{-1}(p) \subseteq X$ on which $G$ also acts. That is, if $F_p$ is a fundamental domain for the action of $G$ on $\mathcal{F}_p$, the union of all pairwise intersections $\cup_{g_1 \neq g_2}(g_1 F_p \cap g_2 F_p)$ have measure 0 and $F_p$ and $Gx$ are differentiable manifolds, eventually with boundary, for all $x \in \mathcal{F}_p$. Note that Example 1 satisfies all the above.

We are now able to state and prove our main theorem.

**Theorem 4.** Denote the fiber $\mathcal{F}_p = h_P^{-1}(p)$. Denote the total density on a fiber $\mathcal{F}_p$ by $q(\mathcal{F}_p) = \int_{\mathcal{F}_p} q(x)dx$ and the renormalized density by $q_p(x) = q(x)/q(\mathcal{F}_p)$. Let $k_p(Gx)$ be the total dissent of an orbit on $\mathcal{F}_p$ with the renormalized probability $q_p(x)$. Let $F_p$ be a fundamental domain for the action of $G$ on $\mathcal{F}_p$. Assume that $h$ is incorrectly invariant under $G$ on each fiber of $\mathcal{F}_p$. In other words, $h$ satisfies $h(gx) = h(x)$ for $g \in G$, $x \in \mathcal{F}_p$, but $f(x) \neq f(gx)$ for some $x, g$. Let $P_1 = \{p : \text{Acc}_p(h) \leq 1/2\}$ and $P_2 = \{p : \text{Acc}_p(h) \geq 1/2\}$. Let $Gx^*$ be the orbit with the smallest nonzero total dissent $k(Gx^*)$, i.e., $x^* = \underset{x \in X}{\arg\min} \, k(Gx)$. Then ECE is bounded above by

$$\text{ECE}(h) \leq \frac{1}{2} + \int_0^1 r(p)\left|\frac{1}{2} - p\right|dp - k(Gx^*)\int_{P_2} r(p)dp.$$

*Proof.* Observe that

$$\left|\text{Acc}_p(h) - p\right| = \left|\text{Acc}_p(h) - \frac{1}{2} + \frac{1}{2} - p\right| \leq \left|\text{Acc}_p(h) - \frac{1}{2}\right| + \left|\frac{1}{2} - p\right|.$$

Integrating over $[0, 1]$,

$$\text{ECE}(h) = \int_{p=0}^{p=1} r(p)\left|\text{Acc}_p(h) - p\right|dp \leq \int_{p=0}^{p=1} r(p)\left(\left|\text{Acc}_p(h) - \frac{1}{2}\right|\right)dp + \int_{p=0}^{p=1} r(p)\left|\frac{1}{2} - p\right|dp.$$

Note $P_1$ and $P_2$ partition $[0, 1]$. By definition of $P_1$ and $P_2$,

$$\int_{p=0}^{p=1} r(p)\left(\left|\text{Acc}_p(h) - \frac{1}{2}\right|\right)dp = \int_{P_1} r(p)\left(\frac{1}{2} - \text{Acc}_p(h)\right)dp + \int_{P_2} r(p)\left(\text{Acc}_p(h) - \frac{1}{2}\right)dp \qquad (4)$$

$$= \frac{1}{2}\left(\int_{P_1} r(p)dp - \int_{P_2} r(p)dp\right) - \int_{P_1} r(p)\text{Acc}_p(h)dp + \int_{P_2} r(p)\text{Acc}_p(h)dp. \qquad (5)$$

By Theorem 1, the accuracy $\text{Acc}_p(h)$ on any fiber of $p$ is bounded above by $1 - \int_{F_p} k_p(Gx)dx$. Combining this bound with the bounds defining $P_1$ and $P_2$ yields,

$$0 < \text{Acc}_p(h) < \min\left(1 - \int_{F_p} k_p(Gx)dx, \frac{1}{2}\right) \qquad \forall p \in P_1.$$

$$\frac{1}{2} < \text{Acc}_p(h) < \left(1 - \int_{F_p} k_p(Gx)dx\right) \qquad \forall p \in P_2.$$

Observe that the upper bound for ECE is determined by the upper bound of Equation 4 and Equation 5 and is related to the accuracy of $h$ by the last two integrals in Equation 5. In particular, the model $h$ that maximizes ECE satisfies $\text{Acc}_p(h) = 0$ on $P_1$ and $\text{Acc}_p(h) = 1 - \int_{F_p} k(Gx, p)dx$ on $P_2$. Substituting these values into Equation 5 gives upper bound

$$\left[ \frac{1}{2}\left( \int_{P_1} r(p)dp - \int_{P_2} r(p)dp \right) + \int_{P_2} r(p)dp - \int_{P_2} r(p)\int_{F_p} k_p(Gx)dxdp \right] + \int_{p=0}^{p=1} r(p)\left| \frac{1}{2} - p \right|dp$$

which simplifies to

$$\frac{1}{2} + \int_{p=0}^{p=1} r(p)\left| \frac{1}{2} - p \right|dp - \int_{P_2} r(p)\int_{F_p} k_p(Gx)dxdp.$$

Finally,

$$
\begin{aligned}
-\int_{P_2} r(p)\int_{F_p} k_p(Gx)dxdp &= -\int_{P_2} r(p)\int_{F_p} \min_{y \in Y} \int_{Gx} q_p(z)\mathbb{1}(f(z) \neq y)dzdxdp \\
&\leq -\int_{P_2} r(p)\int_{F_p} \min_{y \in Y} \int_{Gx} q(z)\mathbb{1}(f(z) \neq y)dzdxdp \\
&\leq -\int_{P_2} r(p)\min_{y \in Y} \int_{Gx^*} q(z)\mathbb{1}(f(z) \neq y)dzdp \\
&= -k(Gx^*)\int_{P_2} r(p)dp
\end{aligned}
$$

and so $\text{ECE}(h) \leq \frac{1}{2} + \int_0^1 r(p)|\frac{1}{2} - p|dp - k(Gx^*)\int_{P_2} r(p)dp$. This completes the proof. $\square$

This bound is non-vacuous. That is, the upper bound is tighter than 1 since it accounts for the error caused by incorrect invariance along the subset of fibers where accuracy is at least 50%. By considering the orbit with the lowest total dissent, we can compute an upper bound that is tighter than 1 even without knowing the error lower bound on each fiber, i.e., $\int_{F_p} k_p(Gx)dx$, or the fibers themselves $\mathcal{F}_p$. In other words, if we consider all of the orbits with incorrect invariance, then the ECE upper bound is only as tight as the smallest orbit-wise error bound will allow. Comparing with Theorem 4, we see that, under the invariance assumption, the upper bound decreases by $k(Gx^*)\int_{P_2} r(p)dp$. In other words, the bound is tight when there is nontrivial dissent on high mass orbits within fibers where the accuracy is greater than or equal to $1/2$. Conversely, the bound becomes trivial for models with correct invariance, or if the partition $P_1$ is a much larger subset than $P_2$.

One tradeoff this bound makes is that it is in terms of $k(Gx^*)$, which only considers error along one orbit. If we know which data points are in each fiber of $h_P$, then we can tighten the bound.

**Corollary 1.** Define $m = \min_{p \in [0,1]} \int_{F_p} k_p(Gx)$. Then $\text{ECE}(h) \leq \frac{1}{2} + \int_0^1 r(p)|\frac{1}{2} - p|dp - m\int_{P_2} r(p)dp$.

*Proof.* We compute $\frac{1}{2} + \int_0^1 r(p)|\frac{1}{2} - p|dp - \int_{P_2} r(p)\int_{F_p} k_p(Gx)dxdp \leq \frac{1}{2} + \int_0^1 r(p)|\frac{1}{2} - p|dp - m\int_{P_2} r(p)dp$. $\square$

A key subtlety in the proof of Corollary 1 is that $m$ is a minimum over error lower bounds defined on fibers of $[0,1]$ and not orbits. This is stated formally in Remark 1.

**Remark 1.** By assumption of invariance on $h_P$, the fibers of $[0,1]$ contain entire orbits. The integrated total dissent $\int_{F_p} k_p(Gx)dx$ is defined on the collection of orbits where the confidence is always given by $h_p(x_p) = p$, but the label $h_Y(x_p)$ itself may vary. This is possible because points $x_{p_1}$ and $x_{p_2}$ may belong to distinct orbits which map to different distinct labels $y_1$ and $y_2$ under $h_Y$ but map to the same confidence $p$ under $h_P$.

We note that for the special case of binary classification, an accuracy of 50% is the minimum accuracy on each fiber. If the accuracy of a classifier is less than 50% on each fiber, we can construct a classifier that simply chooses the opposite label to improve its accuracy so that it is accurate over 50% of the time.

**Corollary 2.** Assume $|Y| = 2$. ECE is bounded above by $1 - k(Gx^*)$. In the special case when we can compute $m = \min\limits_{p \in [0,1]} \int_{F_p} k_p(Gx)$, ECE is bounded above by $1 - m$.

*Proof.* Note that $\int_0^1 r(p)|\frac{1}{2} - p|dp$ is bounded above by $\frac{1}{2}$. We have $\int_{P_2} r(p)dp = \int_0^1 r(p)dp = 1$ by assumption. Substituting these values into Theorem 4 and Corollary 1 completes the proof. $\square$

### 4.3 Improved Bounds for bi-Lipschitz Invariant Functions

Notice that the upper bounds in Proposition 1 and Theorem 4 are in terms of $r(p)$ but this density is not in general easily derivable from $q(x)$. In order to express each bound in terms of $q(x)$, we introduce extra assumptions on $h_P$.

From Hörmander (2015), we have that if $h_P(x)$ is continuously differentiable and has gradient nowhere 0, then

$$r(p) = \int_X q(x)\delta(p - h_P(x))dx = \int_{\mathcal{F}_p} \frac{1}{|\nabla h_P(x)|}q(x)dx_p \tag{6}$$

where $\delta$ is the Dirac-Delta distribution.

To attain an upper bound on ECE independent of $h$, we find an upper bound on $\frac{1}{|\nabla h_P(x)|}$. This is achievable if $h_P(x)$ is bi-Lipschitz, as defined below.

**Definition 4.** Given metric spaces $(X, d_X)$ and $(Y, d_Y)$, a function $f : X \to Y$ is *(upper) Lipschitz continuous* if there exists a constant $K \geq 0$ such that for all $x_1, x_2 \in X$,

$$d_Y(f(x_1), f(x_2)) \leq K d_X(x_1, x_2).$$

Furthermore, a function is $(K_1, K_2)$-*bi-Lipschitz continuous* if it is lower Lipschitz and upper Lipschitz, i.e.,

$$\frac{1}{K_2}d_X(x_1, x_2) \leq d_Y(f(x_1), f(x_2)) \leq K_1 d_X(x_1, x_2).$$

Taking the limit $x_1 \to x_2$ shows the Lipschitz constant $K$ bounds the gradient of $f$. If the function is bi-Lipschitz, then $K_1$ bounds the gradient and $K_2$ bounds the reciprocal of the gradient. In practice, these Lipschitz constants may be very large for arbitrary neural networks, but can be controlled by architectural considerations such as spectral normalization (e.g. Behrmann et al., 2019; Chen et al., 2019).

**Proposition 2.** Assume $h_P(x)$ is differentiable, has gradient nowhere 0, and is $(K_1, K_2)$ bi-Lipschitz continuous. Let $Gx^\diamond$ be the orbit with the least integrated probability density, i.e., $x^\diamond = \arg\min\limits_{x \in X} \int_{Gx} q(x)dx$. Then

$$\text{ECE}(h) \leq \frac{1}{2} + \frac{K_2}{4} + \min\left\{0, -k(Gx^*)K_2 \int_{P_2} dp \int_{Gx^\diamond} q(x)dx\right\}.$$

*Proof.* We start with the bound from Theorem 4. Substituting the expression for $r(p)$ from Equation 6 into the upper bound from Theorem 4 and using the Lipschitz constant to bound the gradient gives

$$\text{ECE}(h) \leq \frac{1}{2} + \int_0^1 \int_{\mathcal{F}_p} K_2 q(x)dx_p \left|\frac{1}{2} - p\right| dp - k(Gx^*) \int_{P_2} \int_{\mathcal{F}_p} K_2 q(x)dx_p dp.$$

We now relate this inequality to integrals over $X$ and $Gx^\diamond$ in order to remove the dependence on $h_p$. Notice,

$$\int_0^1 \int_{\mathcal{F}_p} K_2 q(x)dx_p \left|\frac{1}{2} - p\right| dp \leq \int_0^1 \int_X K_2 q(x)dx \left|\frac{1}{2} - p\right| dp = \frac{K_2}{4}.$$

and

$$\int_{P_2} \int_{\mathcal{F}_p} K_2 q(x)dx_p dp \geq \int_{P_2} \int_{Gx^\diamond} K_2 q(x)dx dp = K_2 \int_{P_2} dp \int_{Gx^\diamond} q(x)dx.$$

Thus,

$$\text{ECE}(h) \leq \frac{1}{2} + \frac{K_2}{4} - k(Gx^*)K_2 \int_{P_2} dp \int_{Gx^\diamond} q(x)dx.$$

The upper bound for Proposition 1 can be derived in the same way or seen as the special case where $k(Gx^*) = 0$ or $P_2 = \emptyset$ and the improvement from Theorem 4 is vacuous. □

As in Corollary 2, the $\int_{P_2} dp$ term is unity in the case of binary classification. This enables us to further characterize the utility of the bound in the examples that follow.

### 4.4 Invariant Upper Bounds Examples

A natural question is how much tighter is the upper bound on ECE under the assumption of invariance (Theorem 4) than the bound in the unconstrained case (Proposition 1). In the case $h_P(x)$ is bi-Lipschitz with constants $K_1$ and $K_2$, Proposition 2 gives a relatively concrete answer; the gap is $k(Gx^*)K_2 \int_{P_2} dp \int_{Gx^\diamond} q(x)dx$. In order to understand the gap when the function is not assumed to be bi-Lipshitz, we consider several examples with specific $r(p)$. The examples illuminate how the tendency of a model to be uncertain can tighten the bound for functions both with and without invariance. Specifically, Example 2 considers a binary classification task where $r(p)$ is a truncated normal distribution. We consider means $\mu$ that correspond to low, medium, and high confidence.

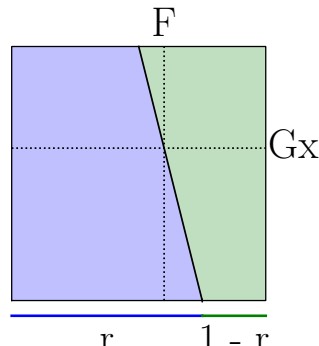

Figure 1: Binary Classification on a Unit Square with Translation Invariance. Blue and green represent true labels.

**Example 2** (Upper Bound Comparison for Binary Classification on the Unit Square). Let $X = \mathbb{R}^2$ with density $p(x, y) = 1$ if $0 \leq x \leq 1$ and $0 \leq y \leq 1$ and $p(x, y) = 0$ otherwise. The function $h$ is invariant to translations in the $x$−direction.

Let us now consider the unconstrained bound for three different example distributions $r(p)$, each corresponding to low, medium, and high confidence.

Recall that the truncated normal distribution with mean $\mu$, variance $\sigma^2$, and bounds $(a, b)$ has a probability density

$$f(x; \mu, \sigma, a, b) = \frac{1}{\sigma} \frac{\varphi\left(\frac{x-\mu}{\sigma}\right)}{\Phi\left(\frac{b-\mu}{\sigma}\right) - \Phi\left(\frac{a-\mu}{\sigma}\right)}$$

for $a \leq x \leq b$ and $f(x) = 0$ otherwise, where

$$\varphi(\xi) = \frac{1}{\sqrt{2\pi}}e^{-\frac{1}{2}\xi^2}, \qquad \Phi(z) = \frac{1}{2}\left(1 + \text{erf}\left(\frac{z}{\sqrt{2}}\right)\right).$$

Set $\sigma = 0.1, a = 0$, and $b = 1$. For $\mu = 0.5$, the ECE upper bound is $\approx 0.58$ by Theorem 4. For $\mu = 0.25$ or for $\mu = 0.75$ the bound is $\approx 0.75$.

Let us now consider the bound constrained by invariance with the same Truncated Normal Densities. As seen in Figure 1, the orbit with the smallest integrated total dissent is the one on the $x$−axis, $k(Gx^*) = 1 - r$.[3] Since this task is binary classification, we have $P_2 = [0, 1]$ and $\int_{P_2} r(p)dp = 1$. The upper bound on ECE, using the assumption of incorrect invariance, decreases by $(1 - r)$ regardless of the mean of $r(p)$.

For functions with and without invariance, we see that the upper bound is tighter when the confidence is concentrated around 50%, which can be interpreted as the model "hedging its bets." That is, the model minimizes calibration error by outputting a confidence value close to the mean possible value. For invariant functions, the bound is always tightened proportionally to the orbit with the highest accuracy, regardless of the distribution $r(p)$.

---

[3]Integrated density over the line segment $[0, 1] \subset [0, 1] \times [0, 1]$ requires more care if we are reasoning about the measures directly, but can be made precise using the disintegration theorem (Pachl, 1978)

The following example illustrates how to use the bound in Corollary 2 and how tight it is. That is, we are in the special case when we know which elements $x$ are in a given fiber $\mathcal{F}_p$ and the task is binary classification.

**Example 3** (Binary Classification with Reflection Invariance)**.** Consider the unit circle $S^1$ embedded in $\mathbb{R}^2$. Along $S^1$ we have 20 points that are each assigned either a blue or orange label. The cyclic group $C_2$ acts on elements of $S^1$ by reflecting them over the $x$-axis and trivially on the labels. As shown in Figure 2, each half of the circle contains at least one orbit with incorrect invariance, though the model is still able to correctly classify 90% of the data on the right half. On the right half, where $x > 0$, assume $h_p$ has confidence $p_1$ on each of the 5 orbits. Assume $h_p$ has confidence $p_2$ on each of the 5 orbits where $x < 0$.

If $p_1 \neq p_2$, then there are two fibers $F_{p_1}$ and $F_{p_2}$. (The fiber $F_{p_1}$ is shown with diagonal lines in Figure 2a.) On the left half, the error $\int_{F_p} k_p(Gx)dx \geq 0.5$, and on the right half, $\int_{F_p} k_p(Gx)dx \geq 0.1$. Taking the minimum and invoking Corollary 2, we find that $m = 0.1$ and ECE is bounded above by $1 - 0.1 = 0.9$.

If $p_1 = p_2$, then there is only one fiber $F_{p_1}$, as in Figure 2b. The error over the whole dataset is bounded from below by $0.5(0.1 + 0.5) = 0.3$, and by Corollary 2 ECE is bounded above by $1 - 0.3 = 0.7$.

Having only two confidence values may be reasonable in real world settings if there is a prevailing noise (e.g., shadow, camera artifacts, background patterns) on just one side of the field of view of a camera, leading to approximately two different confidence regions in our model output. This setup is also conceptually similar to what we will see in Section 4.7, where we will show experimentally how incorrect reflection invariance affects model calibration.

**Example 4** (Binary Classification with Rotation Invariance)**.** Consider the same dataset as in Example 3, but assume $h$ has rotation invariance to $SO(2)$ instead of reflection invariance. In this case, there is only one orbit and so $h_Y$ and $h_P$ each take one value. There is only one fiber $F_p = X$, which is the entire dataset, as illustrated in Figure 2c. The classification error is minimized when $h$ predicts each label as blue, since there are more blue labels then orange labels in our dataset. This gives an error lower bound of 0.3, and an ECE upper bound of 0.7. This is the same result for reflection invariance in the special case where the confidences are the same across each fiber.

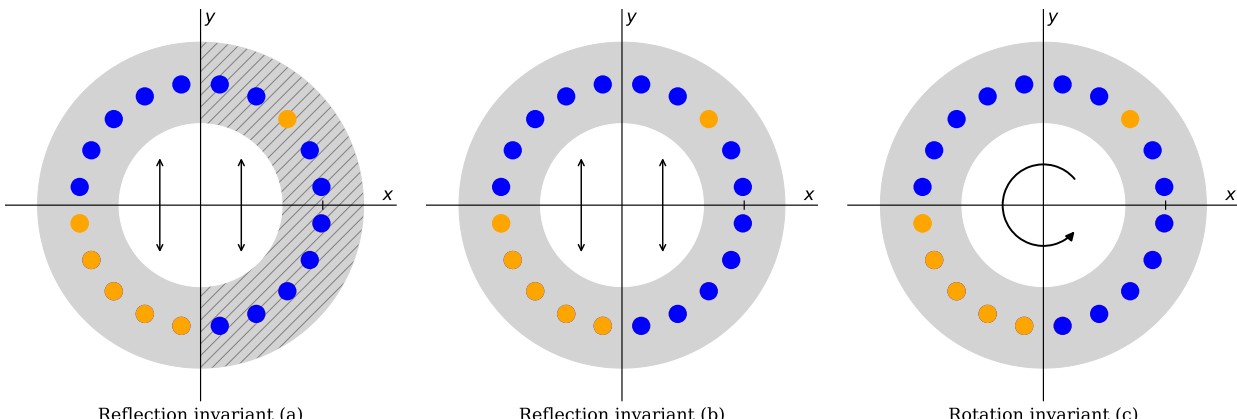

| Reflection invariant (a) | Reflection invariant (b) | Rotation invariant (c) |

Figure 2: Dataset with pointwise incorrect invariance. The first two panels indicate that the model is invariant to the action of $C_2$, which performs reflections over the $x$-axis. The tick mark indicates the $x = 1$ point. The presence of diagonal lines in the first panel depicts that the model has 2 confidence fibers. The last panel indicates that the model has rotation invariance. Colors represent labels in all panels.

## 4.5 ECE Lower Bounds

The assumption of invariance can also be used to obtain an ECE lower bound. The trivial lower bound of 0 is obtained when $\mathrm{Acc}_p(h) = p$ for all $p$. However, if $h$ has an accuracy lower bound $m$, then $\mathrm{Acc}_p(h) \neq p$ for $p < m$, resulting in a tighter bound.

In order to derive the ECE lower bound, we first need to introduce a classification error upper bound. This is defined in terms of the minority label, the label that causes the maximal error on a given orbit $Gx$ (analogous to the majority label that minimizes error on a given orbit in Wang et al. (2024)). The error on this orbit is called the *minority label total dissent*.

**Definition 5** (Minority Label Total Dissent)**.** For an orbit $Gx$ of $x \in X$, the minority label total dissent $\kappa(Gx)$ is the integrated probability density of the elements in the orbit $Gx$ having a different label than the minority label:

$$\kappa(Gx) = \max_{y \in Y} \int_{Gx} q(z) \mathbb{1}(f(z) \neq y)) dz.$$

We prove in Proposition 3 that the total classification error is bounded above by the integrated minority label total dissent.

**Proposition 3.** The classification error is bounded above $\mathrm{err}_{\mathrm{cls}}(h) \leq \int_F \kappa(Gx)dx$. Equivalently, the accuracy is bounded below by $1 - \int_F \kappa(Gx)dx$.

*Proof.* We compute

$$\begin{aligned}
\mathrm{err}_{\mathrm{cls}}(h) &= \int_X q(x) \mathbb{1}(f(x) \neq h(x)) dx \\
&= \int_F \int_{Gx} q(z) \mathbb{1}(f(z) \neq h(z)) dz dx \\
&\leq \int_F \max_{y \in Y} \int_{Gx} q(z) \mathbb{1}(f(z) \neq y)) dx = \int_F \kappa(Gx) dx.
\end{aligned}$$

$\square$

**Note 1.** The bound in Proposition 3 is vacuous if all orbits omit at least one label. In that case, the total minority dissent $\kappa(Gx)$ for each orbit is 0.

**Example 5.** The task is to predict 1 of 12 classes for a series of 12 points distributed along $S^1$. There is a bijection between classes and the data points. Assume a rotation invariant classification model which can only output one class for all the points. Here, the minority label can be any 1 of the 12 labels, resulting in an error of $1/12$.[4]

We use the accuracy lower bound from Proposition 3 to give a lower bound on the ECE.

**Theorem 5.** Denote the fundamental domain of $G$ in $\mathcal{F}_p$ as $F_p$, where $\mathcal{F}_p$ is as defined in Theorem 4. As in Theorem 4, the total minority dissent on an orbit in a fiber $\mathcal{F}_p$ is denoted $\kappa_p(Gx)$ and is defined in terms of the renormalized density $q_p(x) = q(x)/\int_{\mathcal{F}_p} q(x)dx$. Define the minimum fiber-wise classification accuracy as $m = \min_{p \in [0,1]} \left(1 - \int_{F_p} \kappa_p(Gx)dx\right)$. Then ECE is bounded below by $\int_0^m r(p)(m-p)dp$.

*Proof.* By Proposition 3, the classification accuracy on each fiber is bounded below by $1 - \int_{F_p} \kappa_p(Gx)dx$. $\mathrm{Acc}_p(h)$ is therefore bounded below by $m$. Thus $\int_0^1 r(p)|\mathrm{Acc}_p(h) - p|dp \geq \int_0^m r(p)|\mathrm{Acc}_p(h) - p|dp \geq \int_0^m r(p)(m-p)dp$ since $\mathrm{Acc}_p(h) \geq m > p$ by integrating over $0 \leq p \leq m$. $\square$

Recall that for ECE to be well defined, we assume $r(p)$ is nonzero on $[0,1]$. Therefore, $r(p) \neq 0$ anywhere on $[0, m]$ and the lower bound is strictly greater than 0.

We give an accuracy lower bound $m'$ that does not depend on the fibers $\mathcal{F}_p$, which implicitly depend on $h_P$. Later, we will use this bound to express an ECE lower bound which is independent of $h_P$.

**Proposition 4.** Assume that $h_P$ is a continuously differentiable function and that its gradient is nowhere 0. Define $x^* = \underset{x \in X}{\arg\min} \int_{z \in Gx} q(z)dz$ so that the orbit with the smallest integrated density is $Gx^*$. Let

---

[4]i.e., a broken clock is right twice a day

$p^* = \arg\min_{p \in [0,1]} (1 - \int_{F_p} \kappa_p(Gx)dx)$ and let $m' = 1 - \frac{1}{\int_{Gx^*} q(z)dz} \int_F \kappa(Gx)dx$. Define $m$ as in Theorem 5. ECE is bounded below by $\int_0^{m'} \int_{\mathcal{F}_p} \frac{1}{|\nabla h_P(x)|} q(x)dx_p (m' - p)dp$.

*Proof.* As before, we have that $r(p) = \int_X q(x)\delta(p - h_P(x))dx = \int_{\mathcal{F}_p} \frac{1}{|\nabla h_P(x)|} q(x)dx_p$. As in Theorem 4, we note that the accuracy lower bound for each fiber must be computed in terms of the renormalized probabilities. By definition we have

$$m = \min_{p \in [0,1]} 1 - \int_{F_p} \kappa_p(Gx)dx = 1 - \int_{F_p^*} \kappa_p(Gx)dx.$$

Next, we factor out the normalization constant on $\mathcal{F}_{p^*}$ and compare it to the integrated density on the orbit $Gx^*$. This avoids a dependence on $h_P$ in the bound.

$$1 - \int_{F_p^*} \kappa_p(Gx)dx = 1 - \int_{F_p^*} \max_{y \in Y} \int_{Gx} \frac{q(z)}{\int_{\mathcal{F}_{p^*}} q(z)dz} \mathbb{1}(f(z) \neq y)dzdx$$
$$\geq 1 - \frac{1}{\int_{Gx^*} q(z)dz} \int_{F_p^*} \kappa(Gx)dx.$$

Noting that $\int_{F_p^*} \kappa(Gx)dx \leq \int_F \kappa(Gx)dx$ and recalling the definition of $m'$ shows that $m \geq m'$. So, ECE is bounded below by $\int_0^{m'} \int_{\mathcal{F}_p} \frac{1}{|\nabla h_P(x)|} q(x)dx_p (m' - p)dp$. □

This proof indicates that the accuracy lower bound on the entire dataset, inversely weighted by the integrated probability of the least likely orbit, is less than the accuracy lower bound on any given fiber. This allows us to derive an accuracy lower bound $m'$ independent of the fibers $\mathcal{F}_p$.

As with the upper bound in Proposition 2, the lower bound on ECE is related to how quickly the function $h_P$ changes as a function of $x$. We can get a precise lower bound independent of $|\nabla h_P|$ if we have knowledge of the Lipschitz constant.

**Proposition 5.** Assume $h_P$ is differentiable, has gradient nowhere 0, and has Lipschitz constant $K$. Define $Gx^*$ and $m'$ as in Proposition 4. Then, ECE is bounded below by

$$\int_0^{m'} \int_{Gx^*} \frac{q(x)}{K} dx(m' - p)dp.$$

*Proof.* The ECE lower bound from Proposition 4 is minimized when $\frac{1}{|\nabla h_P(x)|}$ is minimized, so we are interested in when $|\nabla h_P(x)|$ is maximized. The upper bound on $|\nabla h_P(x)|$ is given by the Lipschitz constant $K$. Recall that $\int_{\mathcal{F}_p} \frac{1}{K} q(x)dx_P = \int_{F_p} \int_{z \in Gx} q(z) \frac{1}{K} dzdx_P$. Let $Gx_p^*$ be the orbit with the smallest integrated probability density in $F_p$. Then $\int_{F_p} \int_{z \in Gx} q(z) \frac{1}{K} dzdx_P \geq \int_{z \in Gx_p^*} q(z) \frac{1}{K} dz_P$. Now, we remove the dependence on $p$ by considering the orbit with the smallest integrated probability density, including orbits not in $F_p$. This gives us $\int_{z \in Gx_p^*} q(z) \frac{1}{K} dz_p \geq \int_{Gx^*} q(x) \frac{1}{K} dx$. Therefore,

$$\text{ECE}(h) \geq \int_0^{m'} \int_{z \in Gx^*} \frac{q(x)}{K} dx(m' - p)dp.$$

□

## 4.6 Invariant Lower Bound Example

We now apply Proposition 5 to a Lipschitz and invariant network to give an example of a precise lower bound independent of $h_P$.

**Example 6** (ECE lower bound for a shallow DeepSets network). We consider a two-layer network $h_p$ that is permutation-invariant. In particular, we study a modified version of DeepSets (Zaheer et al., 2017), which is designed to process unordered collections such as point clouds by enforcing permutation equivariance.

An input configuration of $n$ elements with $d$-dimensional features is represented by a matrix $A \in \mathbb{R}^{n \times d}$, where each row encodes the features of a single element. Permuting the input corresponds to permuting the rows of $A$, that is, acting on the first index.

This example considers a dataset $X$ with $n = 4!$ points that are generated by permuting the following set of points: $\{\vec{a}, \vec{b}, \vec{c}, \vec{d}\}$. There is only one orbit in this setting, and all permutations of the set are equally probable under $q(x)$. The co-domain has labels $Y = \{0, 1\}$. The ground truth $f$ is a function of the rows of $A$, denoted $A_i$:

$$f(A) = \begin{cases} 0, & A_1 = \vec{a}, \\ 1, & A_1 \neq \vec{a}. \end{cases}$$

To process such data, we employ modified DeepSets-style linear layers of the form

$$W = \tanh(\lambda_1)I + \tanh(\lambda_2)\, 11^\top,$$

where $\lambda_1, \lambda_2 \in \mathbb{R}$ are learnable parameters, $I$ is the $n \times n$ identity matrix, and $1 = (1, \ldots, 1)^\top \in \mathbb{R}^n$. These layers act on data matrices $A \in \mathbb{R}^{n \times d}$, followed by the ReLU nonlinearity.

The construction ensures permutation equivariance, namely for any permutation matrix $P_\pi \in S_n$ we have

$$W \cdot (P_\pi A) = P_\pi \cdot (WA).$$

To get an invariant output, we use a final readout layer of the form $\tanh(\lambda_3)1^T$ where $\lambda_3 \in \mathbb{R}$ is also a learnable parameter. Multiplication with $\lambda_3 1^T$ performs mean pooling over the set dimension. All together,

$$h_P(x) = \tanh(\lambda_3)1^T \text{ReLU}((\tanh(\lambda_1)I + \tanh(\lambda_2)11^T))x. \tag{7}$$

For Lipschitz functions $f$ and $g$ with Lipschitz constants $L_1$ and $L_2$, the composition $f(g(x))$ has Lipschitz constant $L_2 L_1$ and the sum $f(x) + g(x)$ has Lipschitz constant $L_1 + L_2$. Moreover, the Lipschitz constant for a linear map is given by its maximum singular value $\sigma_{\max}$. Finally, note that ReLU is $1-$Lipschitz. Thus, the Lipschitz constant for Equation 7 is $\sigma_{\max}(1^T)(\sigma_{\max}(I) + \sigma_{\max}(11^T)) = \sigma_{\max}(1^T)(1 + \sigma_{\max}(11^T))$. For $n = 24$, we can compute these values to be 24 and approximately 4.9 respectively.

Applying Proposition 5 the lower bound on ECE is approximately

$$\int_0^{m'} \int_{Gx^*} 0.03 \cdot q(x)dx(m' - p)dp.$$

Thus far, we have derived a lower bound that does not depend on the size of the fibers and has no dependence on $|\nabla h_P|$. What is left is to calculate the integrated density on $Gx^*$ and the accuracy lower bound $m'$.

Since there is only one orbit, $Gx^* = X$ and the integrated density over $Gx^*$ is 1. Moreover, $m'$ reduces to the global accuracy lower bound. Noting that $A_1 = \vec{a}$ will occur with probability 0.25 gives us our accuracy lower bound $m'$. So, we compute

$$\int_0^{m'} \int_{Gx^*} 0.03 \cdot q(x)dx(m' - p)dp = \int_0^{0.25} 0.03(0.25 - p)dp = 0.001.$$

Ultimately, the looseness of the bound here is due to the fact that the Lipschitz constant grows linearly with the number of points $n$.

## 4.7 Classification Experiments

### 4.7.1 Swiss Rolls Depict Harmful Invariance.

We experimentally show how incorrect invariance can cause ECE to increase. While we do not necessarily expect this to happen all the time, here we construct a synthetic dataset designed to show this effect is

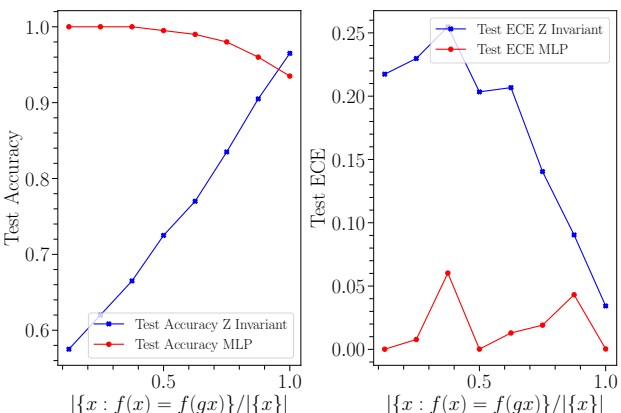

Figure 3: The left plot shows test accuracy for the z-invariant network (blue) and baseline unconstrained MLP (red) under different ratios of correct/incorrect ratios, ranging from 0% to 100% correctness. The right hand plot is the same but for ECE instead of accuracy. As the correct ratio increases, the z-invariant MLP increases in test accuracy and decreases in test ECE, whereas the baseline MLP is relatively flat.

possible. Analogous to how an individual data point can be adversarial to a model, this *dataset* is adversarial to the entire *function space $\mathscr{H}$*.

**Experiment.** This dataset consists of a family of separated Swiss rolls from Wang et al. (2024) with varying levels of correct and incorrect invariance with respect to $z$-translation. These distributions contain a $3D$ point cloud arranged in a spiral-like fashion, and binary labels are assigned to each point. The Swiss rolls have distinct $z$-values that are easily separable with a horizontal plane. See Figure 10 (Appendix C) and Figures 7 and 11 in Wang et al. (2024). We train an unconstrained MLP and a $z$-invariant network to predict the label. We provide further experimental details in Appendix C. Wang et al. (2024) demonstrate a linear increase in test accuracy as a function of correct invariance, and our aim is to realize a similar trend for ECE.

**Results.** Our experiment gives an example where incorrect equivariance harms not only model accuracy, but also model calibration. When the proportion of correct equivariance is low, Figure 3 shows that the model is correct less than about 70% of the time and that ECE may be as high as 25%.

### 4.7.2 Galaxy Zoo Morphology and Uninformative Symmetry Priors.

While the previous experiment showed how a model's ECE can decrease proportionally to the amount of correct invariance in a domain with synthesized data, here we show an example using real data that demonstrates how symmetry can serve as an uninformative prior for improving model calibration.

**Experiment.** We consider the challenging task of galaxy morphology classification. Invariance has shown to improve model accuracy in this domain (Pandya et al., 2023; 2025b). The task naturally has $E(2)$-invariance, where $E(2)$ is the Euclidean group for $\mathbb{R}^2$. We can approximate $E(2)$-invariance in CNNs with $C_n$ and $D_n$ group convolutional layers, where $C_n$ denotes the cyclic group of order $n$ and $D_n$ denotes the dihedral group of order $n$. We examine how ECE changes under stricter symmetry constraints corresponding to higher group order $n$. While any $C_n$ or $D_n$ has correct equivariance, an increase in $n$ certainly captures more of the underlying symmetry as you better approximate the $E(2)$-invariance (with the caveat of aliasing and similar symmetry breaking operations, see Zhang (2019); Karras et al. (2021); Gruver et al. (2023)). We look at trends in both accuracy and ECE under different levels of point-spread function (PSF) convolution. This allows us to look at performance under different levels of ground truth noise, which has the effect of varying both the model confidence and accuracy. PSF blurring is also an extremely relevant source of noise for the astrophysics community, specifically in weak lensing analysis and exoplanet imaging, see Appendix D.1. Our data comes from Galaxy Zoo images (Walmsley et al., 2024) of galaxies from the DESI and SDSS surveys. Further experimental details are recorded in Appendix D.2.

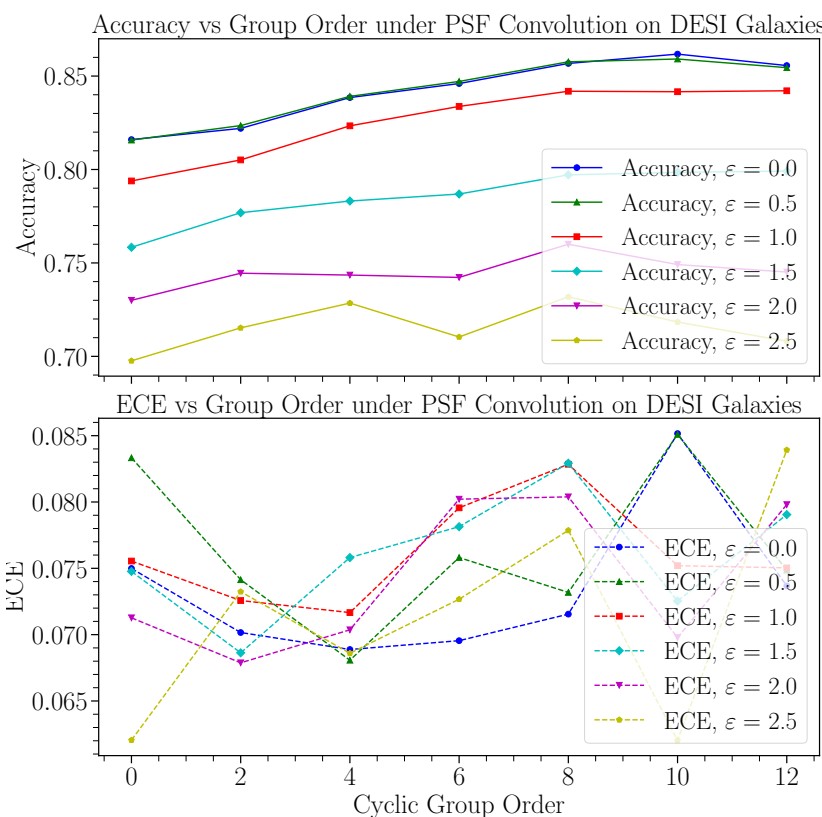

Figure 4: Accuracy and ECE vs Cyclic Group Order under PSF Convolution on DESI Galaxies. The accuracy increases as a function of order especially with low PSF noise, however, the ECE varies tremendously.

**Results.** Figure 4 shows that ECE does not follow a roughly monotonic improvement as we increase the cyclic group order in the same way accuracy does. We interpret these results in the context of the last experiment. While incorrect equivariance can cause a model to become poorly calibrated, that does not imply that correct equivariance provides a direct benefit to model calibration in the same way it does accuracy. This result is echoed in Appendix F, which shows similar trends for both cyclic and dihedral group order for both DESI and SDSS surveys. We note that the accuracy curves are reminiscent of Weiler & Cesa (2019); Pandya et al. (2023) with the asymptotic accuracy increase as a function of group order.

## 5 Invariant and Equivariant Regression Calibration

We now study calibration error in the case of invariant regression. In the process, we define a novel notion of calibration error that works for vector-valued functions. We prove that calibration error is bounded above by an expression that is analogous to the maximum $\chi^2$ error one can obtain averaged over all of the fibers of the uncertainty predictions. The upper bound is determined by the orbit variance of the ground truth on each fiber. We also prove that calibration error is bounded below by the minimum error over all confidence fibers in the setting where the model outputs a univariate Gaussian, which is again determined by the variance of the ground truth on each fiber. Readers interested in experimental consequences of equivariance on uncertainty in the regression setting may skip to Section 6.

### 5.1 Invariant Regression Problem Setup

Consider a function $f : X \to Y$ where $Y = \mathbb{R}^n$. Define a function space $\mathscr{H}$ as the set $\mathscr{H} = \{h \colon X \to \mathcal{M} \times \mathcal{S}\}$. Each function in the space outputs multivariate Gaussian distributions with diagonal covariance. Here,

$\mathcal{M} = \mathbb{R}^n$ represents the space of all mean-vectors and $\mathcal{S} = \mathbb{R}^n_+$ represents the space of all variance-vectors. Denote the two outputs by $h_\mu$ and $h_{\sigma^2}$ and let $p\colon X \to \mathbb{R}$ be a probability density over the domain $X$. Denote the subdomain of $X$ given by the constraint $h_{\sigma^2}(x) = s$ as $X_s = \{x \in X \mid h_{\sigma^2}(x) = s\}$. Denote the fundamental domain of $G$ in $X_s$ as $F_s$. Recall smoothness and separability hypothesis in Assumption 2, which we assume holds for $X_s$ and $F_s$.

Next, we define a family of probability densities for each fiber of $h_{\sigma^2}$. Define a density over $X_s$ by $q_s\colon X \to \mathbb{R}$ via $q_s(x) = \frac{p(x_s)}{\int_{X_s} p(x)dx}$. For $x \notin X_s$, we assume $q_s(x) = 0$. This allows us to define the domain restricted regression error

$$\mathrm{err}_{\mathrm{reg}}(h, s) = \int_{X_s} q_s(x_s)\Big\|h_\mu(x_s) - f(x_s)\Big\|_2^2 dx_s. \tag{8}$$

Denote by $q_s(Gx_s) = \int_{z \in Gx_s} q_s(z)dz$ the probability of the orbit $Gx_s$ on $X_s$. Denote by $q_{\mathrm{norm}}(z) = \frac{q_s(z)}{q_s(Gx_s)}$ the normalized probability density on the orbit $Gx_s$ such that $\int_{Gx_s} q_{\mathrm{norm}}(z)dz = 1$. Let $\mathbb{E}_{Gx_s}[f]$ be the mean of the function $f$ on the orbit $Gx_s$, and let $\mathbb{V}_{Gx_s}[f]$ be the variance of $f$ on the orbit $Gx_s$,

$$\mathbb{E}_{Gx_s}[f] = \int_{Gx_s} q_{\mathrm{norm}}(z)f(z)dz = \frac{\int_{Gx_s} q(z)f(z)dz}{\int_{Gx_s} q(z)dz},$$
$$\mathbb{V}_{Gx_s}[f] = \int_{Gx_s} q_{\mathrm{norm}}(x)\Big\|\mathbb{E}_{Gx_s}[f] - f(z)\Big\|_2^2 dz.$$

These definitions are analogous to those used in deriving the error lower bound on invariant regression in Wang et al. (2024) but restricted to various subsets $X_s$. Intuitively, if a model $h$ is constant on an orbit where $f$ is varying, then the constant that minimizes the regression error is the average of $f$ on the orbit, and the resulting error is the variance of $f$.

Finally, we define a generalization of the expected normalized calibration error (ENCE), as given by Equation 8 in Levi et al. (2022a). There are two main drawbacks of the original ENCE metric: it is defined in terms of binning approximations and assumes that $h_\mu(x)$ and $h_{\sigma^2}(x)$ are scalar values. Our definition not only works for vectors but also avoids discretization, allowing for a discussion of continuous group symmetries. While binning approximations are still necessary to compute ENCE in practice, our theory supports the more generalized continuous case. For estimating ENCE on real datasets, the use of binning approximations is still necessary to estimate $r(s)$. We note that it is advisable to use a numerically stable binning scheme, such as one that considers the quantiles of $s$. In high dimensions, we caution that these bins may be very sparsely populated.

We denote the absolute value of a vector $|s| = (|s_i|)_i$ to be applied element-wise. Similarly, $\sqrt{s}$ denotes the elementwise application of the square root to $s$. Vectors have a partial ordering where $\mathbf{a} \le \mathbf{b}$ if $a_i \le b_i$ for all $i$. Let $D$ be the region of vectors $d$ bounded by $\mathbf{s_1} \le \mathbf{d} \le \mathbf{s_2}$. Define a probability density $r\colon \mathcal{S} \to \mathbb{R}$ such that $\mathbb{P}(h_{\sigma^2}(x) \in D) = \mathbb{P}(s \in D) = \int_D r(s)ds$. This is the push-forward of the density of $p$ over $h_{\sigma^2}$. We assume an analogous condition to Assumption 1: The input $X$ is equipped with an $|X|$ dimensional Hausdorff measure $\mathcal{H}$ so that the push-forward density $r(s)$ is defined with respect to the push-forward measure $h_{\sigma^2}\#(\mathcal{H})$. We note that these conditions do not need to hold for datasets with discrete input and output spaces, which we will explore in a later example.

**Generalized Expected Normalized Calibration Error.** We now have the necessary machinery to define our novel Generalized ENCE (GENCE) metric that applies to vector-valued functions. The goal for our learning task is that $h_\mu$ fits the function $f$ *and* $h_{\sigma^2}$ properly predicts confidence by minimizing GENCE (Equation 9):

**Definition 6** (GENCE)**.** Under a well-specified Gaussian with diagonal covariance, the GENCE metric penalizes the fiber-wise discrepancies between the error and the uncertainty averaged over all variances.

This quantity is defined:

$$\text{GENCE}(h) = \lim_{\varepsilon \to \vec{0}} \int_{\mathcal{S}} r(s) \cdot \frac{\mathbb{E}_{X_s}\left[\left\|\sqrt{\frac{2}{\pi}s} - |h_\mu(x) - f(x)|\right\|_2^2 \,\Big|\, s - \varepsilon < h_{\sigma^2}(x) < s + \varepsilon\right]}{\left\|\sqrt{\frac{2}{\pi}s}\right\|_2^2} ds. \tag{9}$$

**Remark 2.** We clarify why the normalization constant $\sqrt{2/\pi}$ is included in Definition 6. If a model is well calibrated, then $\sqrt{2s/\pi} = \overline{|h_\mu(x) - f(x)|}$ and $s = \overline{(h_\mu(x) - f(x))^2}$ should both hold. The factor of $\sqrt{2/\pi}$ comes from Geary (1935) and is obtained by integrating the product of a normal distribution with the absolute distance between a function and the mean. In other words, if a function $h_{\sigma^2}$ predicts variance $s$, then we should expect the mean absolute distance between $f$ and $h_\mu$ to be $\sqrt{2s/\pi}$. We choose to penalize discrepancies between $\sqrt{2s/\pi}$ and $|h_\mu(x) - f(x)|$ instead of $s$ and $(h_\mu(x) - f(x))^2$ because expressing the error term in terms of absolute value allows us to later apply a theorem from Wang et al. (2024).

As indicated by Remark 2, a potential alternative definition for GENCE is,

$$\text{GENCE}_{\text{sq}}(h) = \lim_{\varepsilon \to \vec{0}} \int_{\mathcal{S}} r(s) \cdot \frac{\mathbb{E}_{X_s}\left[\left\|s - (h_\mu(x) - f(x))^2\right\|_2^2 \,\Big|\, s - \varepsilon < h_{\sigma^2}(x) < s + \varepsilon\right]}{\left\|s\right\|_2^2} ds \tag{10}$$

where $(\cdot)^2$ is applied element-wise. Unless otherwise stated, we will use the version of GENCE specified in Equation 9. We further distinguish by using $\text{GENCE}(h)$ for Equation 9 and $\text{GENCE}_{\text{sq}}(h)$ for Equation 10.

As we did for classification, we will omit the limit $\varepsilon \to 0$ in Equation 9 and Equation 10 for brevity, but continue to assume the expression is well defined when $r(s) \neq 0$ and $X$ has a Hausdorff measure $\mathcal{H}$. We assume that the function space $\mathscr{H}$ is arbitrarily expressive except that it is constrained to be invariant with respect to a group $G$. That is to say, it is a universal approximator for compactly supported $G$-invariant functions.

We adopt $\text{GENCE}(h)$ and $\text{GENCE}_{\text{sq}}(h)$ as calibration metrics because it enables us to continue our strategy of bounding calibration errors by noting that the calibration function is equivalent to a fiber-wise error, but note that alternative frameworks such as coverage based metrics are also compelling. In particular, extending Dobriban & Yu (2025) to cases of symmetry mismatch would serve as a complement to this work.

## 5.2 GENCE Invariant Upper Bound

We now state the first theorem of this section, which bounds GENCE for $G$-invariant functions and discusses the special case of minimized regression error using the bounds from Wang et al. (2024). This bound can be used to interpret how poorly calibrated an invariant model can be when it is performing near its optimal capability in terms of regression error.

**Theorem 6.** If a model $h$ is $G$-invariant, then GENCE as defined in Equation 9 is bounded as follows

$$0 \leq \text{GENCE}(h) \leq 1 + \mathbb{E}_{\mathcal{S}}\left[\frac{\text{err}_{\text{reg}}(h, s)}{\|\sqrt{\frac{2}{\pi}}s\|_2^2}\right].$$

If $\text{err}_{\text{reg}}(h, s)$ is minimized, then ENCE is bounded by

$$0 \leq \text{GENCE}(h) \leq 1 + \mathbb{E}_{\mathcal{S}}\left[\frac{\int_{F_s} q_s(Gx_s)\mathbb{V}_{Gx_s}[f]dx_s}{\|\sqrt{\frac{2}{\pi}}s\|_2^2}\right].$$

*Proof.* By the triangle inequality we have that

$$0 \leq \mathbb{E}_{X_s}\left[\left\|\sqrt{\frac{2}{\pi}}s - |h_\mu(x) - f(x)|\right\|_2^2 \,\Big|\, h_{\sigma^2}(x) = s\right] \leq \mathbb{E}_{X_s}\left[\left\|\sqrt{\frac{2}{\pi}}s\right\|_2^2 + \left\||h_\mu(x) - f(x)|\right\|_2^2 \,\Big|\, h_{\sigma^2}(x) = s\right].$$

Consequently,

$$0 \leq \text{GENCE}(h) \leq \int_{\mathcal{S}} r(s) \cdot \frac{\mathbb{E}_{X_s}\left[\|\sqrt{\frac{2}{\pi}s}\|_2^2 + \||h_\mu(x) - f(x)\||_2^2 \mid h_{\sigma^2}(x) = s\right]}{\left\|\sqrt{\frac{2}{\pi}s}\right\|_2^2} ds$$

$$= \int_{\mathcal{S}} r(s) \cdot \left[1 + \frac{\int_{X_s} q_s(x_s)\left\|h_\mu(x) - f(x)\right\|_2^2 dx}{\left\|\sqrt{\frac{2}{\pi}s}\right\|_2^2}\right] ds.$$

By the definition of domain restricted regression error in our problem setup we have

$$\text{GENCE}(h) \leq \int_{\mathcal{S}} r(s) \left[1 + \frac{\text{err}_{\text{reg}}(h,s)}{\left\|\sqrt{\frac{2}{\pi}s}\right\|_2^2}\right] ds = 1 + \mathbb{E}_{\mathcal{S}}\left[\frac{\text{err}_{\text{reg}}(h,s)}{\left\|\sqrt{\frac{2}{\pi}s}\right\|_2^2}\right]. \tag{11}$$

Now, if the domain restricted regression error is minimized, then by [Theorem 2](#) we have that

$$\text{err}_{\text{reg}}(h,s) = \int_{F_s} q_s(Gx_s)\mathbb{V}_{Gx_s}[f]dx_s \tag{12}$$

which completes the proof. □

### 5.3 GENCE Equivariant Upper Bound

We now generalize the bounds on calibration error for invariant models ([Theorem 6](#)) to *equivariant* models. While the results are similar to the invariant case, the proof strategy is different. This is because the fibers are no longer closed under the action of the group. That is, $G$ acting on $X_s \subseteq X$ may result in a vector in $X \setminus X_s$. Similar to [Theorem 6](#), our bound can be used to interpret how poorly calibrated an *equivariant* model can be when it is performing near its optimal capability in terms of regression error.

We assume similar notation and hypotheses as in invariant regression, with one additional modification. We will treat the matrix $Q_{Gx}$ from [Theorem 3](#) as a function of $s$. In particular, we replace the probability density $p(x)$ with the fiber-wise renormalized probabilities $q_s(x)$. Recalling that $q_s(x)$ assigns no measure to elements in $X \setminus X_s$, we can apply [Theorem 3](#) to bound the regression error on individual fibers.

**Theorem 7.** Assume the function space $\mathscr{H}$ is equivariant. GENCE is bounded as follows

$$0 \leq \text{GENCE}(h) \leq 1 + \mathbb{E}_{\mathcal{S}}\left[\frac{\text{err}_{\text{reg}}(h,s)}{\|\sqrt{\frac{2}{\pi}s}\|_2^2}\right].$$

If $\text{err}_{\text{reg}}(h,s)$ is minimized then GENCE is bounded by

$$0 \leq \text{GENCE}(h) \leq 1 + \mathbb{E}_{\mathcal{S}}\left[\frac{\int_F \int_G q(gx)\|f(gx) - g\mathcal{E}_G[f,x]\|_2^2 \alpha(x,g)dgdx}{\|\sqrt{\frac{2}{\pi}s}\|_2^2}\right].$$

Unlike before, we can not decompose an integral over $X_s$ into an iterated integral over $F_s$ and $Gx_s$. This is because while $G$-invariance of $h_{\sigma^2}$ implies that the fibers $X_s$ are closed under the action of $G$, this does not hold when $h_{\sigma^2}$ is *equivariant.* The proof strategy is instead to relate the domain restricted error to the error over the entire domain.

*Proof of [Theorem 7](#).* Consider the integral in [Equation 8](#). Since $q_s(x) = 0$ for $h_{\sigma^2}(x) \neq s$, we may write

$$\text{err}_{\text{reg}}(h,s) = \int_X q_s(x)\left\|h_\mu(x) - f(x)\right\|_2^2 dx.$$

Using the same argument as we did with invariant regression (Theorem 6), we arrive at

$$0 \leq \mathrm{GENCE}(h) \leq \int_{\mathcal{S}} r(s) \left[ 1 + \frac{\mathrm{err}_{\mathrm{reg}}(h,s)}{\|\sqrt{\frac{2}{\pi}}s\|_2^2} \right] ds = 1 + \mathop{\mathbb{E}}_{\mathcal{S}} \left[ \frac{\mathrm{err}_{\mathrm{reg}}(h,s)}{\|\sqrt{\frac{2}{\pi}}s\|_2^2} \right].$$

Applying Theorem 3, if $\mathrm{err}_{\mathrm{reg}}(h,s)$ is a minimizer then we have

$$0 \leq \mathrm{GENCE}(h) \leq 1 + \mathop{\mathbb{E}}_{\mathcal{S}} \left[ \frac{\int_F \int_G q(gx) \|f(gx) - g\mathcal{E}_G[f,x]\|_2^2 \alpha(x,g) dg dx}{\|\sqrt{\frac{2}{\pi}}s\|_2^2} \right]. \tag{13}$$

This completes the proof. □

The upper bounds presented in this section are analogous to the maximum $\chi^2$ score one can attain averaged over all of the variances the model predicts. Consequently, fibers $\mathcal{F}_s$ where the regression error is small may contribute more to the upper bound than other fibers if the variance $s$ is also smaller. That being said, this $\chi^2$ value may not exhibit favorable convergence properties.

**Remark 3.** The upper bounds in Equation 12 and Equation 13 are not guaranteed to converge to a finite value since $\mathrm{err}_{\mathrm{reg}}(h,s)$ may increase as $s$ goes to infinity. It is tempting to suggest that $r(s)$ has bounded support (and therefore the integral converges). However, $r(s)$ must be supported everywhere in order for GENCE to be well defined. This suggests that bounds on GENCE are only finite when the regression error exhibits nice convergence properties, such as $\lim_{s \to \infty} \mathrm{err}_{\mathrm{reg}}(h,s) < \log s$. Some works (e.g., Vaicenavicius et al., 2019) circumnavigate this by defining the calibration error as the integrated density only on the set $\{s \in S : r(s) \neq 0\}$, i.e., the set where calibration error meets a necessary well-definedness property.

Remark 3 elucidates a key difference between the classification bounds and regression bounds. For classification, ECE is bounded on $[0,1]$ because it is defined as the average of a bounded random variable. In contrast, both the error and uncertainty in regression can become arbitrarily large or small. The assumption of equivariance controls only how small we can make the error, but not the uncertainty.

In practice, it is difficult to compute the upper bound (Theorem 6 and Theorem 7) precisely without prior knowledge on the density of the input data or the density of the push-forward. The true bound can be estimated for trained networks using sample predictions on the data to approximate $r(s)$, however, this may become intractable when the Gaussian is high dimensional. This is because the number of required samples to estimate the probability mass $r(s)$ on a small subset of $\mathbb{R}^n$ grows quickly with $n$. Alternatively, $r(s)$ can be analytically derived when the input distribution is well behaved and the model is simple. For example, a linear model with invertible weights will map a Gaussian distribution $p(x)$ to another Gaussian with a new mean and covariance. Despite these complications, we can consider special cases where the bound can be computed exactly.

### 5.4 GENCE Invariant Lower Bound

In the scalar case, where both the space of mean vectors $\mathcal{M}$ and the space of variance vectors $\mathcal{S}$ are one dimensional, we establish a lower bound on $\mathrm{GENCE}_{\mathrm{sq}}(h)$ using the same strategy as for ECE in Theorem 5. The bound can be computed in a straightforward manner analogously to the classification lower bound, and the special cases for Lipschitz functions still apply. Moreover, the cases where the bound becomes vacuous are more apparent and again are analogous to what was proved for Theorem 5.

**Theorem 8.** Assume that the elements of the function space $\mathscr{H}$ are invariant under the action of $G$ on $X$ and $\mathcal{M} = \mathbb{R}$, $\mathcal{S} = \mathbb{R}_+$. Denote by $m$ the error lower bound, $m = \min_{s \in \mathbb{R}_+} \int_{F_s} q_s(Gx_s) \mathbb{V}_{Gx_s}[f] dx$. Then, $\mathrm{GENCE}_{\mathrm{sq}}(h)$ given by Equation 10 is bounded below by $\int_0^m \frac{r(s)}{s^2}(s-m)^2 ds$.

*Proof.* We start by simplifying the $\mathrm{GENCE}_{\mathrm{sq}}(h)$ expression for scalar mean and variance,

$$\mathrm{GENCE}_{\mathrm{sq}}(h) = \int_0^\infty \frac{r(s)}{s^2} \cdot \mathbb{E}_{X_s} \left[ \left( s - (h_\mu(x) - f(x))^2 \right)^2 \mid h_{\sigma^2}(x) = s \right] ds.$$

Applying Jensen's inequality gives

$$\mathrm{GENCE_{sq}}(h) \geq \int_0^\infty \frac{r(s)}{s^2} \cdot \left( \mathbb{E}_{X_s} \left[ s - (h_\mu(x) - f(x))^2 \,\middle|\, h_{\sigma^2}(x) = s \right] \right)^2 ds.$$

By linearity of expectation we have

$$\int_0^\infty \frac{r(s)}{s^2} \cdot \left( \mathbb{E}_{X_s} \left[ s - (h_\mu(x) - f(x))^2 \,\middle|\, h_{\sigma^2}(x) = s \right] \right)^2 ds = \int_0^\infty \frac{r(s)}{s^2} \cdot \left( s - \mathrm{err_{reg}}(h,s) \right)^2 ds.$$

Applying Theorem 2 and the definition of $m$ then gives the lower bound

$$\int_0^\infty \frac{r(s)}{s^2} \cdot \left( s - \mathrm{err_{reg}}(h,s) \right)^2 ds \geq \int_0^m \frac{r(s)}{s^2} \cdot (s - m)^2 ds$$

since $\mathrm{err_{reg}}(h,s) \geq m > s$ by integrating over $m \geq s \geq 0$. $\qquad\square$

Analogous results to Proposition 4 and Proposition 5 can be obtained for regression under assumption of Lipschitz continuity. Our theorem and proof strategy show that the classification and regression calibration lower bounds are both related to invariance by the minimum fiber-wise approximation error bounds when the outputs in $\mathcal{M}$ and $\mathcal{S}$ are scalar valued. Similar to Note 1, this bound is vacuous only under the condition that $f$ is constant on each of the orbits on one of the fibers, in which case $m = 0$.

### 5.5 Example of GENCE Upper Bound for Invariant Models

**Example 7.** The goal of this example is to show how the GENCE bound can be computed in a small synthetic example and to illustrate how the bound can be tighter or looser depending on the norm of the variances. In this example, we consider a set $X$ of five point clouds. We calculate the GENCE upper bound assuming a minimized regression error. We assume that the function space $\mathcal{H}$ has $E(2)$-invariance. As in Section 4.7, this *dataset* is adversarial to the entire *function space $\mathcal{H}$* given that the ground truth function $f$ is not $E(2)$-invariant, as indicated in Figure 5. The dataset contains some duplicates of the point clouds up to transformations in $E(2)$. Different point cloud orientations are notated with $+$ and $\times$.

In particular, we label the point clouds in $X$ as $a(+), a(\times), b, c,$ and $d$. The corresponding probabilities of obtaining each point cloud are $0.125, 0.125, 0.125, 0.125, 0.5$. This defines the discrete probability distribution $p$ of the input data on $X$. The ground truth function $f \colon X \to Y$ yields four distinct outputs. Any approximation of $f$ by an $E(2)$-invariant model $h$ produces only three distinct lines, since it identifies the outputs of $b$ and $c$, which lie in the same $E(2)$-orbit in this example. The model $h$ produces two distinct variance vectors; it predicts variance $s_1$ for the first two orbits and variance $s_2$ for the last orbit. For visualization purposes we assume these variances to be constant on each dimension, but note this need not be the case.

We now compute the upper bound by computing the regression error lower bound along each fiber. We start with the first fiber (rows 1 and 2 in Figure 5). Since $h$ is $E(2)$-invariant, $h$ is able to fully fit the function $f$ on the first orbit (row 1) containing two rotated versions of $a$. Thus, the regression error is zero. For the second row, $b$ and $c$ are equally probable, and the output of $h$ that minimizes the regression error is just the average of the two lines. To compute the error on the second orbit (row 2), we also need to compute the integrated density:

$$q(Gx_s) = \frac{p(b) + p(c)}{p(a(+)) + p(a(\times)) + p(b) + p(c)} = 0.5.$$

Applying Equation 12 gives $\mathrm{err_{reg}}(h, s_1) = 0.5 \cdot \mathbb{V}_{Gx_{s_1}}[f] = \pi/8$. Now, as with the first orbit, the minimizing regression error on the third orbit (row 3) is zero, since there is only one element in $X$ we can fit. So, $\mathrm{err_{reg}}(h, s_2) = 0$. Finally, that the probability of sampling an element on the fiber of $s_1$ is the same as the probability of sampling an element on $s_2$. Thus, by Theorem 2,

$$\mathrm{GENCE}(h) \leq 1 + \left( 0.5 \cdot \frac{0}{\|\sqrt{\frac{2}{\pi} s_2}\|_2^2} \right) + \left( 0.5 \cdot \frac{\pi/8}{\|\sqrt{\frac{2}{\pi} s_1}\|_2^2} \right) = 1 + \frac{\pi/8}{\|\sqrt{\frac{2}{\pi} s_1}\|_2^2}.$$

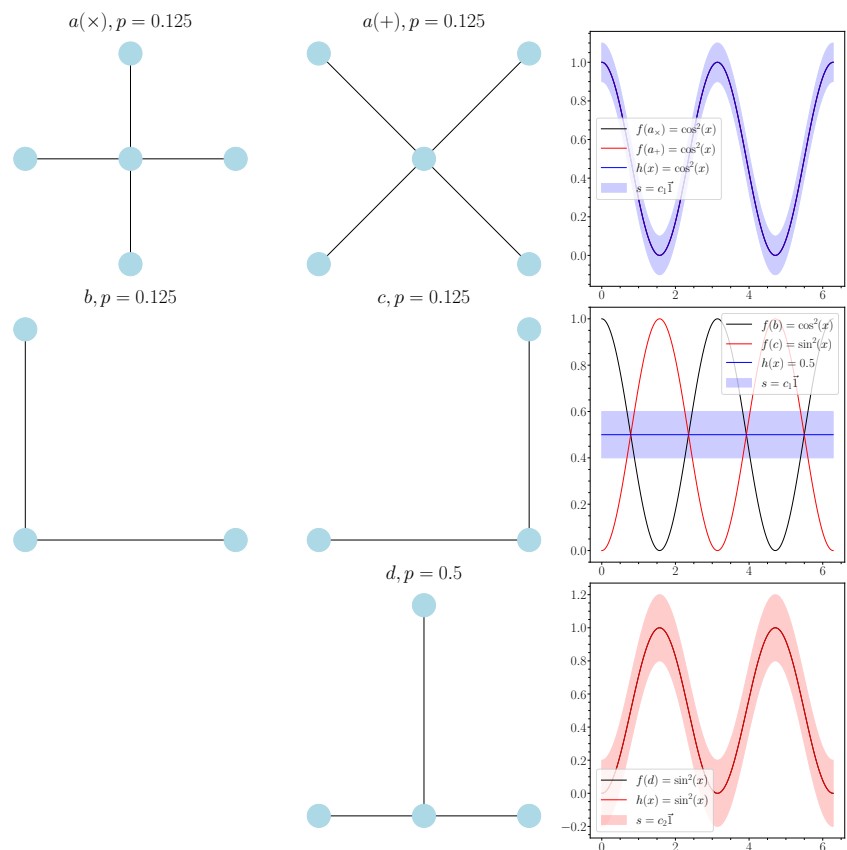

Figure 5: An example on how the ENCE upper bound behaves for an $E(2)$-invariant model $h$ on a set of point clouds and output space of vectors in $\mathbb{R}^n$, depicted as 1D-lines. Each row shows an orbit from the representative set of $X$ in the first two columns as well as the lines produced by $f$ and $h_\mu$ in the third column, with the corresponding variance $s$ predicted by $h_{\sigma^2}(x)$ indicated by the ribbon. Each point cloud is titled with its name in the set $X$ and the probability of sampling it in $X$. It is noteworthy that the orbits in rows 1 and 2 are in the same confidence fiber, and only the second row has a nontrivial regression error lower bound for invariant models.

The upper bound for GENCE is thus $1 + \frac{\pi/8}{\|\sqrt{\frac{2}{\pi}s_1}\|_2^2}$. We can see that in the limit as $\|\sqrt{\frac{2}{\pi}s_1}\|_2^2$ goes to infinity, the upper bound on GENCE becomes 1. Alternatively, in the limit as $\|\sqrt{\frac{2}{\pi}s_1}\|_2^2$ goes to 0, the upper bound on GENCE diverges. The interpretation of the latter is that if the model is extremely confident, then any deviations from the mean prediction represents similarly extreme miscalibration.

## 6 Disentangling Aleatoric and Epistemic Uncertainty

This section details how models that are overconstrained by symmetry may misattribute the source of their uncertainties. In the previous sections, we analyzed the effects of equivariance on model miscalibration. However, our analysis did not distinguish between epistemic and aleatoric uncertainty. Understanding the breakdown between the two sources is important because it allows users to determine whether they have reached the noise floor of their problem or their model is inadequate or poorly trained. While there are clear benefits to distinguishing between the two uncertainties, doing so in practice is known to be hard. Techniques such as evidential regression (as outlined in Section 3.5) are far from perfect in their ability to disentangle these uncertainties (Ovadia et al., 2019; Valdenegro-Toro & Mori, 2022; Osband et al., 2023; Wimmer et al., 2023; Nevin et al., 2024; Jürgens et al., 2024).

### 6.1 Preliminaries

With calibration error, the uncertainty predictions are often interpreted to be the epistemic uncertainty and do not consider aleatoric uncertainties. Thus, we now frame the problem of model calibration using a different metric, the *aleatoric bleed*, which we will define after detailing the relevant background from Section 3.5.

Recall the definitions of aleatoric and epistemic uncertainty in Section 3.5, which we repeat here for convenience.

**Aleatoric Uncertainty:** Aleatoric uncertainty refers to the irreducible part of the uncertainty. Given spaces $X$ and $Y$ and an instance $x_q \in X$, the aleatoric uncertainty is the spread in $p(y|x_q)$.

**Epistemic Uncertainty:** Model uncertainty and approximation uncertainty, on the other hand, are subsumed under the notion of epistemic uncertainty. Let the spaces $X$ and $Y$ be the same as before. Let $l : Y \times Y \to \mathbb{R}$ be the loss function and let $f^*$ be the associated point-wise Bayes predictor defined as

$$f^*(x) := \arg\min_{\hat{y} \in Y} \int_Y l(y, \hat{y}) dP(y|x). \tag{14}$$

Epistemic uncertainty is the uncertainty due to the lack of knowledge of the perfect predictor Equation 14.

Denote the ground truth aleatoric uncertainty (the true dispersion) at a given point $x$ by $f(x)$. Now consider an equivariant function space $\mathscr{H} = \{h : X \to \mathcal{S}_{\text{aleatoric}}\}$. We assume both $f$ and elements in $\mathscr{H}$ are nonnegative, and $\mathscr{H}$ is arbitrarily expressive.

**Definition 7.** The *aleatoric bleed* is the regression error $\text{err}_{\text{reg}}(h)$ for a function space $\mathscr{H} = \{h\colon X \to \mathcal{S}_{\text{aleatoric}}\}$.

Put differently, the aleatoric bleed measures the epistemic mass spuriously identified as aleatoric mass. Typical ensemble spread is roughly epistemic only if the aleatoric uncertainty is correctly modeled, otherwise the uncertainties may leak both ways. Aleatoric bleed is a means of quantifying the degree to which epistemic error induced by symmetry mismatch is absorbed by the aleatoric channel of a parametric predictor.

Again for convenience, we now recap the specific parameterization of the uncertainties via evidential regression that was outlined in Section 3.5. Given data points $(y_1, \ldots, y_n) \sim \mathcal{N}(\mu, \sigma^2)$, we may impose priors $\mu \sim \mathcal{N}(\gamma, \sigma^2 \nu^{-1})$, $\sigma^2 \sim \Gamma^{-1}(\alpha, \beta)$ where $\Gamma(\cdot)$ is the gamma function, $m = (\gamma, \nu, \alpha, \beta)$, and $\gamma \in \mathbb{R}$, $\nu > 0$, $\alpha > 1$, $\beta > 0$. One can then show that $p(y_i|m) = St(y_i; \gamma, \beta(1 + \nu)/\alpha\nu, 2\alpha)$, where the St distribution has probability density given by

$$St(t; \mu, \sigma, \nu) = \frac{\Gamma\left(\frac{\nu+1}{2}\right)}{\sqrt{\pi\nu}\sigma\Gamma\left(\frac{\nu}{2}\right)} \left(1 + \frac{1}{\nu}\left(\frac{t-\mu}{\sigma}\right)^2\right)^{-(\nu+1)/2}.$$

Parameterizing the Student's t distribution as a four parameter family is useful because it lends itself to the following definitions of the prediction, aleatoric uncertainty, and epistemic uncertainty:

$$\begin{aligned}
\mathbb{E}[\mu] &= \gamma \quad \text{(Prediction)} \\
\mathbb{E}[\sigma^2] &= \frac{\beta}{\alpha - 1} \quad \text{(Aleatoric Uncertainty)} \\
\text{Var}[\mu] &= \frac{\beta}{\nu(\alpha - 1)} \quad \text{(Epistemic Uncertainty)}.
\end{aligned}$$

If a function $h$ estimates the aleatoric uncertainty using evidential regression, then the aleatoric bleed is the error between predicted uncertainty vectors $\hat{s}$ with elements $\hat{s}_i = \beta_i/(\alpha_i - 1)$ and a ground truth uncertainty $s$. We note that by Theorem 3, the aleatoric bleed has a known lower bound. Our choice of evidential regression as opposed to other uncertainty decompositions such as mutual information is motivated by the issues laid out in Wimmer et al. (2023). While evidential regression can struggle to perform uncertainty

decomposition due to convergence issues (Bengs et al., 2022; Meinert et al., 2023; Jürgens et al., 2024), we expect it to perform better in the cases without symmetry mismatch and with a known ground truth aleatoric uncertainty and ultimately provide a strong baseline performance.

We highlight that this bound is structural (from symmetry) rather than statistical. This is useful because even with correct equivariance, the aleatoric and epistemic separation is not identifiable without repeated-measure or instrument-noise structure. In this setting, the aleatoric uncertainty can be estimated from samples or modeled using domain knowledge, and any residual uncertainty is known to be epistemic. The only thing that can be detected is calibration failures on fibers of predicted variance, which we explored in Section 4 and Section 5. Lastly, we notice that the aleatoric bleed can be easily computed in cases where the ground truth aleatoric uncertainty is identically the zero vector, which we will explore in the experiments that follow.

### 6.2 Aleatoric Bleed Experiments

We now present numerical experiments that highlight how overconstrained models experience increased aleatoric bleed. In particular, we show how incorrect invariance and equivariance can cause an increase in aleatoric bleed, but correct equivariance does not result in reduced bleed compared to unconstrained baselines. Incorrect equivariance serves as a structural source of epistemic error, as it forces the hypothesis class to misrepresent orbit structure. We consider aleatoric bleed for a synthetic vector field prediction task as well two real chemical property prediction tasks. We point out the need for supplemental qualitative analysis when the output space is vector valued.

Our experiments often use evidential regression to estimate the epistemic and aleatoric uncertainties. For convenience, we remind the reader of the key formalisms for evidential regression that were outlined in the background section (Section 3.5) here. The goal of a neural network is to maximize the likelihood of seeing data points $y$ under the Student's t distribution: $St(y_i; \gamma, \frac{\beta(1+\nu)}{\alpha\nu}, 2\alpha)$. The prediction, aleatoric uncertainty, and epistemic uncertainty are then defined by $\gamma$, $\beta/(\alpha-1)$, and $(\beta/\nu) \cdot (1/(\alpha-1))$ respectively.

#### 6.2.1 Vector Field Regression.

This experiment demonstrates an intuitive example where incorrect and extrinsic equivariance contributes to aleatoric bleed. Consider a model $h : \mathbb{R}^3 \to \mathbb{R}^3 \times \mathbb{R}^3$ that predicts two vector fields representing a mean and a variance prediction. That is, we predict two vectors attributed to any given point in $\mathbb{R}^3$ indicative of a prediction and an aleatoric uncertainty. We denote the ground truth vector field at a given point $x$ by $f(x)$, and $h$ is constrained to be $E(3)$-equivariant. Figure 6 presents two examples of how the equivariance taxonomy can result in different levels of aleatoric bleed.

To examine how the equivariance taxonomy influences downstream aleatoric bleed, we consider two different ground truth functions $f$ designed to produce both correct and incorrect $E(3)$-equivariance:

1. ***Spiral.*** $f(x) = Qx$, with $Q$ a 90° rotation matrix in $\mathbb{R}^3$.

2. ***Sinusoidal.*** $f(x) = -\sin^2(||x||)x$, scaling each input by a sinusoidal radial factor.

The spiral dataset contains pointwise incorrect and extrinsic $E(3)$-equivariance, since in general rotations in $\mathbb{R}^3$ do not commute. For the sinusoidal case, we note that rotations, translations, and reflections in $\mathbb{R}^3$ preserve the norm of a vector $x$, which is suffcient to ensure that our network has correct equivariance (Satorras et al., 2021). In both cases, $f$ is completely deterministic, meaning any nonzero variance vector is indicative of aleatoric bleed.

For simplicity and visualization purposes, our dataset consists of vectors in $\mathbb{R}^3$ with a $z$ component of 0, and we choose rotation matrices $Q$ that keep the vectors in the $xy$-plane. We provide relevant training details in Appendix B.

**Results.** As expected, Figure 6 shows that the incorrect and extrinsic equivariance makes the $E(3)-$equivariant model unable to fit the data appropriately with its mean predictions. Consequently,

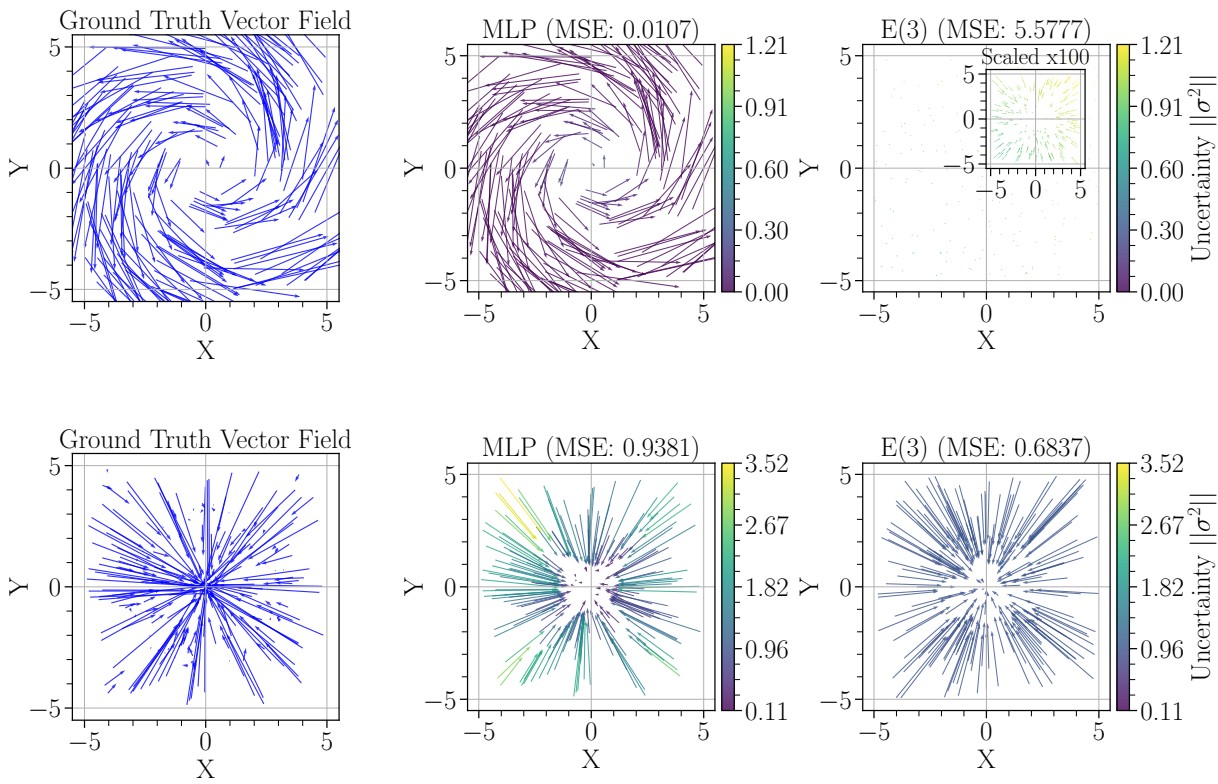

Figure 6: Vector regression results for the rotational and sinusoidal datasets (top and bottom respectively). For the model predictions in the middle and right columns, the color of the vector indicates the norm of the variance. The mean prediction vectors of the $E(3)$-equivariant model on the rotational dataset have very small magnitude. The inset shows them scaled by a factor of 100 for visibility. For the rotational dataset the equivariance is not correct and the E(3) model struggles to predict the vector and compensates with high variance. For the sinusoidal dataset, the equivariance is correct, and the E(3) model suffers from less aleatoric bleed.

it predicts extremely high variance vectors, as our $\beta-$NLL loss function can reach a local minimum when the variance prediction is significantly larger than the mean squared error. As shown in Figure 7, despite the fact that the mean vector field fails to appropriately fit the data, the $\beta-$NLL loss is still fairly close to the MLP for vectors at any given angle. However, in the case of the correct equivariance with the sinusoidal dataset, the correctly applied $E(3)-$equivariance helps the model both in terms of MSE and $\beta-$NLL, accurately fitting the data and minimizing aleatoric bleed. One way to interpret the result is that, because the equivariance condition is misspecified, the learning process pushes residual epistemic uncertainty into the aleatoric uncertainty estimate.

### 6.2.2 Chemical Properties and Aleatoric Bleed

The goal of this experiment is to assess whether a model's learned variance predictions are themselves reliable in a setting that is more realistic than the vector fields in Figure 6. That is, we ask if the model's confidence predictions are consistent with what the ground truth variance should be, and how equivariance can affect this.

**Scalar-Valued Predictions.** This scalar-valued property prediction tasks take as input chemical compounds sourced from QM9s (Ramakrishnan et al., 2014). We predict various chemical properties with two different message-passing graph neural networks, one non-equivariant baseline and one with $E(3)$-equivariance.

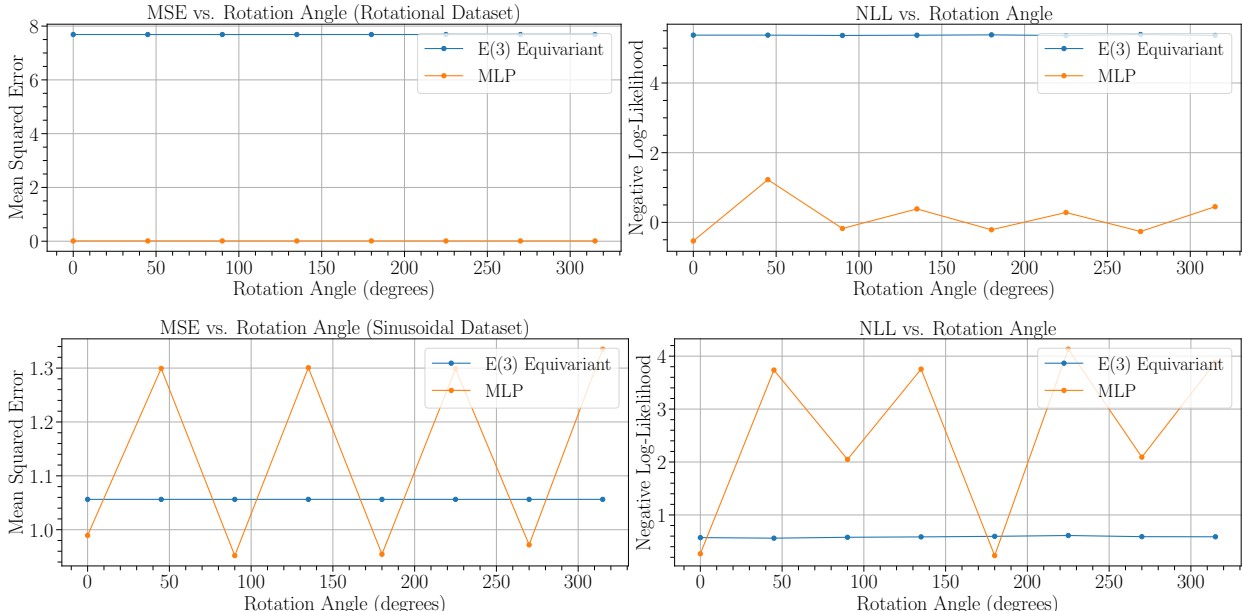

Figure 7: MSE and $\beta$-NLL losses for different rotation angle in the $xy-$plane for the rotational and sinusoidal datasets (top and bottom respectively). In the sinusoidal case, the MSE and NLL loses are constant as a function of the angle in a way that is helpful due to correct equivariance.

Specifically, we employ the GIN model (Xu et al., 2018) as a non-equivariant baseline and compare it with an $E(3)$-invariant model (Batzner et al., 2022), using implementations based on Backenköhler et al. (2023). Both models are equipped with independent feed forward neural network decoder heads which are used to learn a four parameter family that characterizes a Student's t distribution. This in turn gives us enough degrees of freedom to reconcile epistemic and aleatoric uncertainties as described in Section 3.5. Further experimental details are provided in Appendix E. Physically, the relationship between these scalar values and chemical compounds should be a deterministic process, and accordingly the ground truth aleatoric uncertainty should always be zero. Since we are dealing with scalar values, aleatoric bleed reduces from a norm to a simple average over the square of predicted uncertainties. Note that the goal with this experiment is not to train the models to optimal performance; in fact, having models that cannot perfectly generalize is useful for us to study models with non-trivial uncertainties. We strive instead to compare models with similar accuracy but potentially varying levels of aleatoric bleed.

**Scalar-Valued Results.** In contrast to Figure 6, which showed the negative impacts of incorrect and extrinsic equivariance on aleatoric bleed, we find that correct equivariance has little impact on aleatoric bleeding. As a specific case study, consider the dipole moment prediction task shown in Figure 8. We see that the aleatoric bleed is nearly identical between the GIN and $E(3)$-invariant models, with no significant deviations in how errors are distributed. This result mirrors what we found for classification: while correct equivariance provides limited gains in calibration, violations of equivariance can substantially degrade it.

In Table 1, we compare the aleatoric bleed between the baseline and equivariant models when their accuracy is comparable, which we quantify as having a mean absolute error (MAE) within 0.25. The model with the lower aleatoric bleed seems to depend neither on the performance of the model nor the inclusion of equivariance. Our findings support the same conclusion that correct equivariance does little to help prevent aleatoric bleed.

In Section 6, we discussed how the aleatoric bleed has a known lower bound. The results from this experiment suggest that the lower bound is not tight enough to be meaningful to practitioners working on scalar-valued properties in the QM9 dataset. This is likely because the invariance constraint here is correct, so the lower

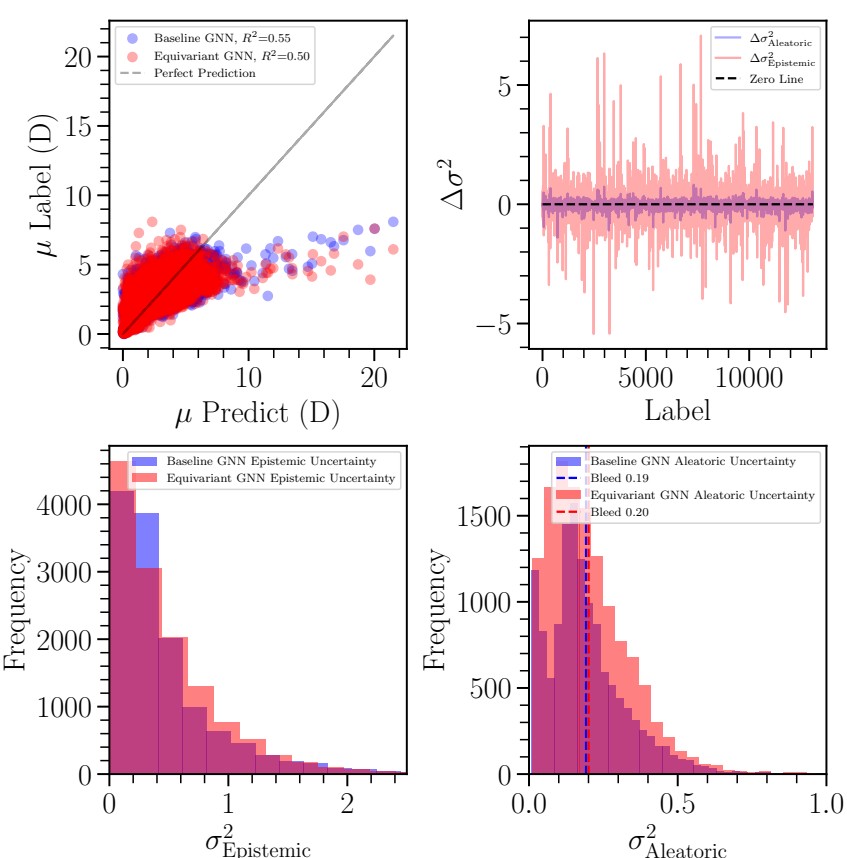

Figure 8: **Top Left:** Prediction versus label for both GIN and $E(3)-$Invariant models. The two models perform similarly in terms of regression. **Top Right:** The difference between aleatoric and epistemic uncertainties between the two models for each label. The model's uncertainty estimates tend to be consistent with one another. **Bottom Left:** A distribution of the epistemic uncertainty predictions for the two models. **Bottom Right:** A distribution of the aleatoric uncertainty predictions for the two models. We see that the distribution for epistemic uncertainty has a fatter tail than the aleatoric uncertainty distribution.

bound on aleatoric bleed is zero. This is in contrast to the vector regression spiral experiment where incorrect and extrinsinc equivariance clearly caused an increase of aleatoric bleed.

| Chemical Property | Unit | GIN MAE | $E(3)-$Invariant MAE | GIN AB | $E(3)-$Invariant AB |
|---|---|---|---|---|---|
| $\varepsilon_{LUMO}$ | eV | 0.4404 | 0.6710 | **0.0081** | 3.0288 |
| $\Delta\varepsilon$ | eV | 0.6877 | 0.7283 | **0.0013** | 3.0287 |
| $U_0$ | eV | 0.2563 | 0.0563 | 0.0333 | **0.0053** |
| $U$ | eV | 0.2558 | 0.0563 | 0.0323 | **0.0053** |
| $U_0^{\text{ATOM}}$ | eV | 0.1908 | 0.1458 | 0.0193 | **0.0052** |
| $G^{\text{ATOM}}$ | eV | 0.7954 | 0.7706 | **0.0000** | 0.0014 |
| $A$ | GHz | 0.0667 | 0.2504 | **0.0000** | 0.0014 |
| $B$ | GHz | 0.1625 | 0.0992 | 0.0066 | **0.0040** |
| $C$ | GHz | 0.0780 | 0.0493 | **0.0008** | 0.0013 |
| $\langle R^2 \rangle$ | $(a_0)^2$ | 0.8621 | 0.8956 | **0.0005** | 0.0013 |

Table 1: Accuracy and Aleatoric Bleed (AB) for various scalar properties in QM9S for baseline and equivariant graph neural network models. Predictions and error estimates are given for the $z-$scored scalar values.

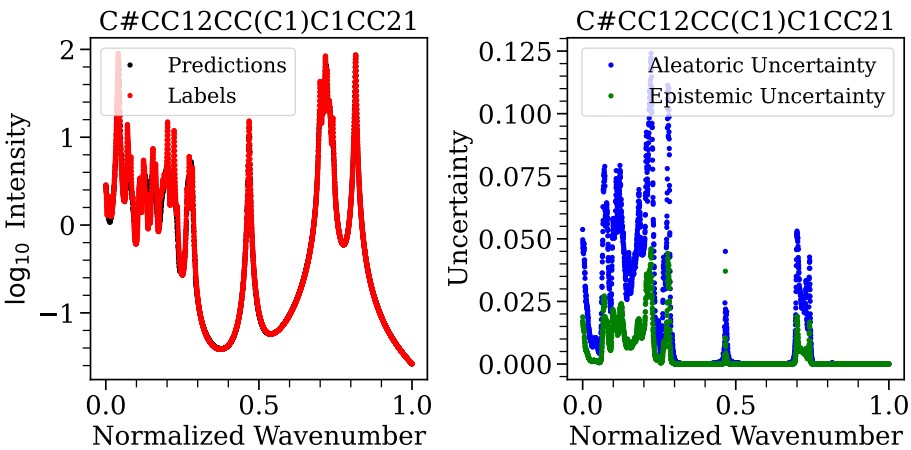

Figure 9: **Left:** Sample prediction vs ground truth spectra for the molecule given by SMILES (Weininger, 1988) string $C\#CC12CC(C1)C1CC21$. **Right:** The model's predicted aleatoric and epistemic uncertainties for each of the normalized wavenumbers.

**Vector-Valued Predictions.** This experiment highlights a need for qualitative analysis to work in tandem with our aleatoric bleed metric for high-dimensional outputs. In particular, aleatoric bleed fails to describe the individual coordinates in which the predicted variance vector suffers the most in terms of bleeding.

This experiment again uses QM9s, however, this time we instead predict spectral lines emitted from the chemical compounds using a network with steerable $E(3)-$vectors (Brandstetter et al., 2021). As before, we use independent feed forward neural network decoder heads in order to learn a four parameter family that characterizes a Student's t-distribution. The ground truth mapping from molecule to label should be deterministic and accordingly the aleatoric uncertainties should be zero. Any non-zero uncertainties are indicative of epistemic uncertainties bleeding into the aleatoric uncertainty prediction. As such, the aleatoric bleed becomes the mean squared norm of the predicted aleatoric variance vectors.

**Vector-Valued Results.** As seen in Figure 9, the model tends to conflate high frequency signals with noise. The model's aleatoric uncertainty follows the epistemic uncertainty quite closely, indicating that the model can not tell them apart. We compute an aleatoric bleed of $\approx 17.613$, however, this does not indicate *where* or *which dimension* contributes to most of the bleed. We conclude that the aleatoric bleed is only able to tell a practitioner that the model is confusing different sources of uncertainty. Evidently, the metric can not tell a user *where* the model tends to poorly estimate the uncertainty estimate without additional diagnostics. We suggest practitioners use aleatoric bleed in conjuction with visual aids like Figure 9 to fully assess the quality of a model's uncertainty estimates.

Additionally, while not our main contribution, we point out that our model's raw performance is comparable to the state-of-the-art DetaNet (Zou et al., 2023), with both models achieving $R^2$ scores above 0.9 and often close to 1.0 on QM9s test molecules. Our steerable $E(3)$ vector-based model also provides initial steps towards quantifying its own uncertainty. We further discuss the merits and limitations of our approach compared to DetaNet in Appendix E. We leave further model development and evaluation as an oppurtunity for future work.

## 7 Limitations

In this section, we outline some known limitations of this study which coincide with directions for future work. The theorems in this work are predicated on the assumption that orbits and fundamental domains are differentiable manifolds (Assumption 2), which may not always be true in practice (Dym et al., 2024). Another limitation is that we need to assume a strong hypothesis, Lipschitz continuity, to express the

bounds in terms of a density on the input domain $X$. This is a limitation because a density on $X$ is what a user is more likely to have access to. Moreover, the practical usefulness of the bounds is dependent on Lipshitz constants $K$, which may be very high for typical neural network architectures. Our study also focused mostly on correct and incorrect equivariance, but further experimental characterization of the effect of extrinsic equivariance and comparisons to covariate shift in general are not fully addressed in this work. On the experimental side, a noteworthy limitation is that ECE and ENCE are difficult to compute in practice due to discrete binning approximations (Pernot, 2023), as the lack of an unbiased estimator adds uncertainty as to how reliable computed ECE and ENCE scores are. This is especially significant in the regime where the variance is very high-dimensional. In our classification experiments, we addressed this by increasing the granularity at which we bin, see Appendix C and Appendix D. ECE as a metric for model calibration is limited in other aspects as well. In particular, ECE only computes miscalibration with respect to a single label, but does not consider secondary or tertiary outputs that may be useful to practitioners as done in Nixon et al. (2019).

## 8 Future Work

Having laid the groundwork for a first theory for equivariance and uncertainty, there are several interesting avenues for future work. An unbiased estimator for ECE and ENCE that does not depend on binning approximations would be an extremely useful contribution. Having such would enable the ability to interpret model calibration on an absolute scale instead of only being able to compare relative performances that are each dependent on discretization biases.

We also note that our proof strategies for bounding calibration error used symmetry constraints on the model class of equivariant functions, however, constraints beyond symmetry may also lend themselves to our approach. Concretely, any constraint that can be used to bound fiber-wise errors can be used to then bound calibration error.

This work may be extended to broader experimental domains, for example, robotics and cosmology. In these areas, there is a clear intersection between uncertainty and equivariance. Uncertainty and equivariance have proved to be indispensable in robotics in particular due to cost of data collection and the safety-critical applications. In robotics, future work will examine calibration error for equivariant models for imitation learning. In particular, our approach is better suited for imitation learning than reinforcement learning. While our work can be difficult to frame in typical Q-learning setups since the optimal solution to the Bellman update function is unique, a discussion of both equivariance and uncertainty quantification lends itself naturally to behavior cloning tasks (Florence et al., 2022).

Another closely related experimental domain is cosmological large-scale structure. In particular, future work may assess calibration error with symmetry-preserving models using the benchmark released in Balla et al. (2024). The appeal of this benchmark is that it includes graph-level predictions on $\Lambda$CDM (Ryden, 2016; Carroll, 2019) cosmological parameters $\Omega_m$ and $\sigma_8$. Moreover, the $\Omega_m$ and $\sigma_8$ parameters are relevant measurements detailing the matter density of our universe. In cosmology, these predictions are more commonly phrased as constraints on posterior distributions rather than point estimates (e.g., Dark Energy Survey Collaboration et al., 2016; Abbott et al., 2022). Therefore, it is reasonable to apply our framework to this dataset and in particular assess if equivariance can help models distinguish between epistemic and aleatoric uncertainties.

## 9 Discussion and Conclusions

Experiments in the natural sciences, especially in data-sparse settings, strongly benefit from both equivariance and uncertainty estimation, and yet, no general theory for explaining how equivariance relates to uncertainty exists in the literature. We fill this gap, presenting the first theory explaining how equivariance relates to uncertainty estimation. We prove both lower and upper bounds on model calibration error for invariant and equivariant model classes. We do this in both classification and regression settings. We confirm and validate the theory in a set of examples and experiments studying the relationship between

equivariance and uncertainty. Moreover, we show that the theoretical results provide intuition and generally match experimental results even when the hypothesis are not strictly satisfied.

The core conclusions are best explained through the lens of model mispecification, and highlight how equivariant neural networks can fail to meet their calibration objectives on datasets that do not share the same symmetries. We highlight how when equivariant model assumptions are violated, i.e., cases of incorrect and extrinsic equivariance where the model is either over-constrained by symmetry or forced to treat out-of-distribution points as in-distribution, model calibration for both classification and regression tasks is provably worse. Our experiments support these conclusions as well. In the cases of the swiss roll and vector field regression experiments, both incorrect and extrinsic equivariance not only make the model less accurate, but also poorly calibrated. In the case of the vector field regression experiment we also saw that incorrect and extrinsic equivariance contributed to aleatoric bleed. By contrast, we have shown that a model with correct equivariance is not necessarily better calibrated than a similarly sized non-equivariant baseline. As illustrated by the galaxy morphology classification and scalar-valued chemical property prediction experiments, equivariance is not strong enough to help a model become well calibrated nor is it strong enough to prevent aleatoric bleeding. It was only for the highly synthetic vector regression experiment on the sinusoidal dataset that the introduction of correct equivariance was able to significantly improve the raw performance and prevent aleatoric bleeding. The vector regression result is especially interesting in light of the galaxy morphology classification experiment, which showed that correct equivariance can help a model perform better without necessarily making it better calibrated.

## 10 Reproducability Statement

Our codebase containing all of our experiments, as well as the instructions to reproduce our results, is publicly available at `github.com/EdwardBerman/EquiUQ`.

We provide further experimental details and sources for the datasets used in this work throughout Appendices B, C, D, and E. We'd like to highlight the work of Wang et al. (2024) and Pandya et al. (2025a); the artifacts associated with these two works were simple to reproduce and aided us in our study.

## 11 Ethics Statement

Authors have no conflicts of interest to disclose.

### Acknowledgments

The authors thank `Shubhendu Trivedi` for thoughtful discourse and detailed review on an initial draft of our manuscript. E.B. and J.G. acknowledge the feedback and overwhelming support from students in Northeastern University's Mathematics Research Capstone course. E.B. additionally thanks the Biomarkers team at AstroAI (Garraffo, 2024). Finally, E.B., J.G., and M.P. thank the members of the `Geometric Learning Lab` for helpful discussions and welcoming support.

R.W. would like to acknowledge support from NSF Grants 2442658 and 2134178. This work is supported by the National Science Foundation under Cooperative Agreement PHY-2019786 (The NSF AI Institute for Artificial Intelligence and Fundamental Interactions, `http://iaifi.org/`).

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

## A    Iterated Integration

If we assume that $\cup_{g_1 \neq g_2}(g_1 F \cap g_2 F)$ has measure 0 and $F$ and $Gx$ are differentiable manifolds, then we may lift an integral $Gx$ to itself. Denote the identification of the orbit $Gx$ and coset space $G/G_x$ with respect to the stabilizer $G_x = \{g \colon gx = x\}$ by $a_x \colon G/G_x \to Gx$. Then we have

$$\int_{Gx} f(z)dz = \int_G f(gx)\alpha(g,x)dg$$

where

$$\alpha(g,x) = \left(\int_{Gx} dh\right)^{-1} \left| \frac{\partial a_x(\bar{g})}{\partial \bar{g}} \right|.$$

## B    Vector Regression Setup

Our training of the $E(3)-$equivariant neural network uses e3nn_jax (Geiger et al., 2022; Geiger & Smidt, 2022; Kondor et al., 2018; Weiler et al., 2018; Thomas et al., 2018). The MLP baseline is built entirely with Flax (Heek et al., 2024). The models are trained for a minimum of 10 epochs for a maximum number of 100, with early stopping if the validation loss stops improving after 5 epochs. We train on 2000 generated samples. We train using both a $\beta-$NLL loss (Seitzer et al., 2022) and an MSE loss, equally weighted, with $\beta = 1$. The $\beta-$NLL loss is given by

$$\mathcal{L}_{\beta-NLL} = \mathbb{E}_{X,Y}\left[\lfloor\hat{\sigma}^{2\beta}\rfloor \left(\frac{1}{2}\log\hat{\sigma}^2 + \frac{(Y - \hat{\mu}(X))^2}{2\hat{\sigma}^2}\right) + C\right]$$

where $\lfloor\cdot\rceil$ represents a stop-gradient. For consistency with the figures, the reported metrics are calculated on the $xy-$coordinates. The MSE and $\beta$-NLL scores are average over all vectors and $xy-$coordinates.

## C    Swiss Roll Experiment Details

The Swiss Roll distributions are created by generating points in polar coordinates using some $r$ as a function of $\theta$. Additionally, the points are given a $z-$coordinate of 0 or 1. An example of a spiral distribution with extrinsic equivariance seen from a $z-$invariant point-of-view is given in Figure 10. See also Figures 7 and 11 in Wang et al. (2024). The correct and incorrect Swiss Roll Distributions are similar. For correct equivariance, the color labels are the same for each spiral at $z = 0$ and $z = 1$. For incorrect, the labels are the opposite. For extrinsic, the spirals do not overlap. For details further, see Wang et al. (2024).

**Binning Approximations.**    We compute ECE using the following binning approximations:

$$\text{acc}(B_m) = \frac{1}{|B_m|}\sum_{i \in B_m} \mathbb{1}(f = h_Y) \tag{15}$$

$$\text{conf}(B_m) = \frac{1}{|B_m|}\sum_{i \in B_m} h_P \tag{16}$$

$$\text{ECE} = \sum_{m=1}^M \frac{|B_m|}{n} \left| \text{acc}(B_m) - \text{conf}(B_m) \right|. \tag{17}$$

We use 100 bins. We adapt models, data generation, and training materials from Wang et al. (2024) and https://github.com/pointW/ext_theory/. The $z-$invariant network is implemented using DSS layers (Maron et al., 2020).

**Sample Calibration Approximation Error.**    Proposition 1 tells us that ECE is bounded on a closed interval (regardless of any assumption of invariance). This allows us to say something about how many samples we need to approximate the true ECE using Hoeffding's Inequality (Hoeffding, 1963; 1994).

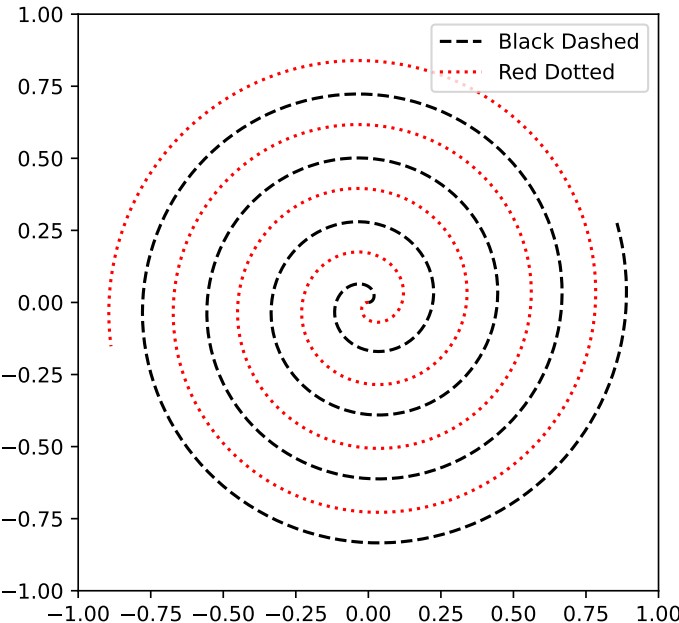

Figure 10: The extrinsic Swiss Roll Distribution seen from a $z-$invariant point of view.

**Proposition 6.** Define the calibration CE as the term inside the integrand of Equation 2, $\text{CE} = |\text{Acc}_p(h)-p|$. For an i.i.d. set of $n$ samples $\{(x_1, h(x_1)), \ldots, (x_n, h(x_n))\}$, we index a given pair by $Z_i$. We have that

$$\mathbb{P}\left[\left|\frac{1}{n}\sum_{i=1}^{n}CE(Z_i) - ECE(Z)\right| > \varepsilon\right] \leq 2\exp(-2n\varepsilon^2)$$

for all $\varepsilon > 0$.

*Proof.* Since Proposition 1 tells us that ECE is bounded on $[0, 1]$, the result follows immediately from Hoeffding's Inequality. $\square$

## D   Galaxy Experiment Details

### D.1   Motivation and Implementation of PSF Blurring

**Motivation.**   A point-spread function (PSF) is an impulse response of an optical system to light. PSFs occur all throughout medical and astronomical imaging. The science case we explore in this work is the distortion of galaxy images. With next generation imagers like JWST and large astronomical surveys like COSMOS-Web (Casey et al., 2023), there are renewed efforts to characterize the effect of the PSF and its effects on downstream scientific analysis (Perrin et al., 2014; Birrer et al., 2021; Jarvis et al., 2021; Michalewicz et al., 2023; Liaudat et al., 2023; Berman et al., 2024; Berman & McCleary, 2024; Feng et al., 2025; Polzin, 2025). Understanding how the PSF harms a model's ability to identify a galaxy's morphology class can hint at the effect of the PSF on measured ellipticity moments (Hirata & Seljak, 2003; Mandelbaum et al., 2005), which is a crucial ingredient for maps of large scale structure (McCleary et al., 2015; 2020; Scognamiglio, 2024).

**Implementation.** The way we implement PSF blurring follows Pandya et al. (2025a). Consider an image grid $I$ with values $(\zeta, \xi)$ and channels $c$. PSF blurring with a Gaussian kernel of width $\epsilon$ via

$$I_{\mathrm{PSF}}(\zeta, \xi) = (I * G)(\zeta, \xi),$$

where

$$G(\zeta, \xi) = \frac{1}{2\pi\epsilon^2} \exp\left(-\frac{\zeta^2 + \xi^2}{2\epsilon^2}\right).$$

We apply this convolution on each channel $c$.

## D.2  Training and Evaluating

Our models and training scripts are adapted from Pandya et al. (2025a) and `https://github.com/deepskies/SIDDA`. The galaxy datasets are initially sourced from `https://zenodo.org/records/14583107`, and there is a script to produce the datasets with PSF blurring in our artifact. We compute ECE using the same approximations as in Equations 15 - 17. We summarize the number of parameters for each model in Table 2 below:

| Model | CNN | $C_2$ | $C_4$ | $C_6$ | $C_8$ | $C_{10}$ | $C_{12}$ |
|---|---|---|---|---|---|---|---|
| Parameters | $1,188,486$ | $1,190,070$ | $1,197,750$ | $1,205,430$ | $1,213,110$ | $1,220,790$ | $1,228,470$ |
| Model | $-$ | $D_2$ | $D_4$ | $D_6$ | $D_8$ | $D_{10}$ | $D_{12}$ |
| Parameters | $-$ | $1,197,750$ | $1,213,110$ | $1,228,470$ | $1,243,830$ | $1,259,190$ | $1,274,550$ |

Table 2: Number of model parameters for Galaxy CNN and GCNN group order experiment.

For further guidance on how many hidden units are needed to approximate the ground truth as a function of group order, we direct the reader to Theorem 16 in Lawrence (2022).

# E  Chemical Property Experiment Details

Our experiment for the chemical properties used a modified version of Backenköhler et al. (2023) for the data preprocessing and main training loop. While their analysis uses one feed forward network head for the prediction task, we use four independent feed forward heads that predict the quantities $m = (\gamma, \nu, \alpha, \beta)$. We train with a negative log likelihood loss function with an added regression loss regularizer,

$$
\begin{aligned}
\Omega &= 2\beta(1+\nu) \\
\mathcal{L}_i^{\mathrm{NLL}}(w) &= \frac{1}{2}\log\left(\frac{\pi}{\nu}\right) - \alpha \log(\Omega) \\
&\quad + \left(\alpha + \frac{1}{2}\right)\log((y_i - \gamma)^2 \nu + \Omega) + \log\left(\frac{\Gamma(\alpha)}{\Gamma(\alpha + \frac{1}{2})}\right) \\
\mathcal{L}_i^{\mathrm{R}}(w) &= |y_i - \mathbb{E}[\mu_i]| \cdot \Phi \\
&= |y_i - \gamma| \cdot (2\nu + \alpha) \\
\mathcal{L}_i(w) &= \mathcal{L}_i^{\mathrm{NLL}}(w) + \lambda \mathcal{L}_i^{\mathrm{R}}(w).
\end{aligned}
$$

The GIN model has $52,417$ paramaters and the $E(3)$-invariant model has $51,969$. Through ablation study, we found that training stability is sensitive to a choice of $\lambda$, which we choose to be either $\lambda = 0.1$ or $\lambda = 1$. This instability is consistent with §S2.1.3 in Amini et al. (2020). Additionally, we found $z-$scoring the training, validation, and testing sets was necessary for ensuring stability during training for all molecular properties outside of the dipole moment.

Our model for emulating spectral lines is trained in the same way, partially taking inspiration from Zou et al. (2023). We note the following tradeoffs between our approach and DetaNet:

1. Our adoption of the message-passing framework is more general than the attentional one used in their work (Bronstein et al., 2021).

2. DetaNet has arbitrary resolution, relying on a sum of basis functions.

3. DetaNet is trained not to produce the spectral line directly, but to produce the dipole moment, polarizability, and the inter-atomic and atomic hessians, which in turn gives the spectral line.

4. DetaNet can be limited by its usage of the Quantum Harmonic Oscillator approximation in some cases.

We leave further compaison and model development as an opportunity for future work. Other potential baselines could include Equiformer (Liao & Smidt, 2022), EquiformerV2 (Liao et al., 2023), Graphormer (Shi et al., 2022), or Graphormer with data augmentation. The model we use in this work has $36,997,125$ parameters.

## F Galaxy Experiment Additional Results

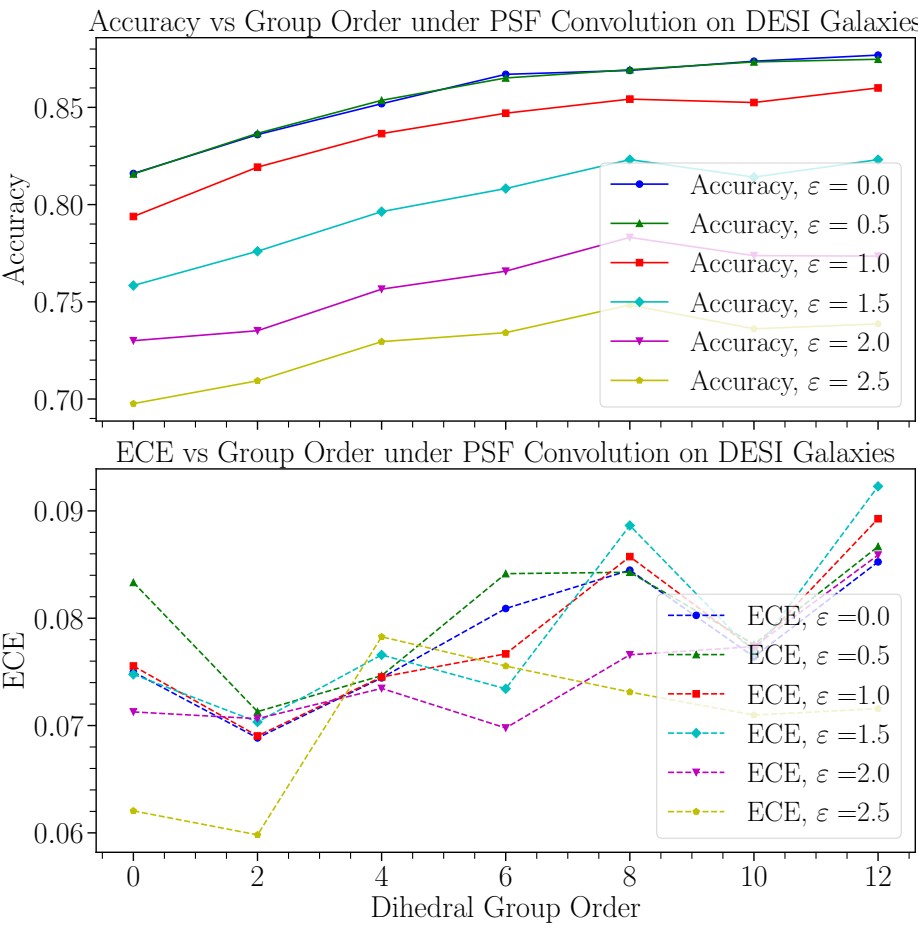

Figure 11: Accuracy and ECE vs Dihedral Group Order under PSF Convolution on DESI Galaxies.

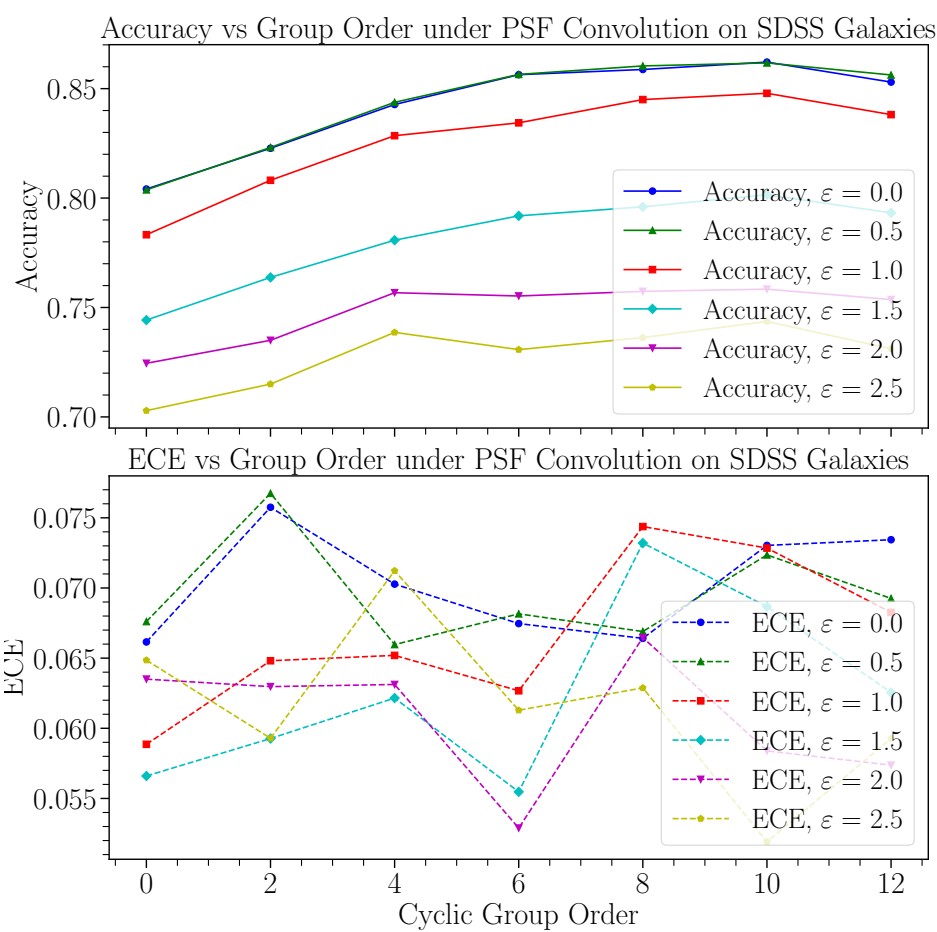

Figure 12: Accuracy and ECE vs Cyclic Group Order under PSF Convolution on SDSS Galaxies.

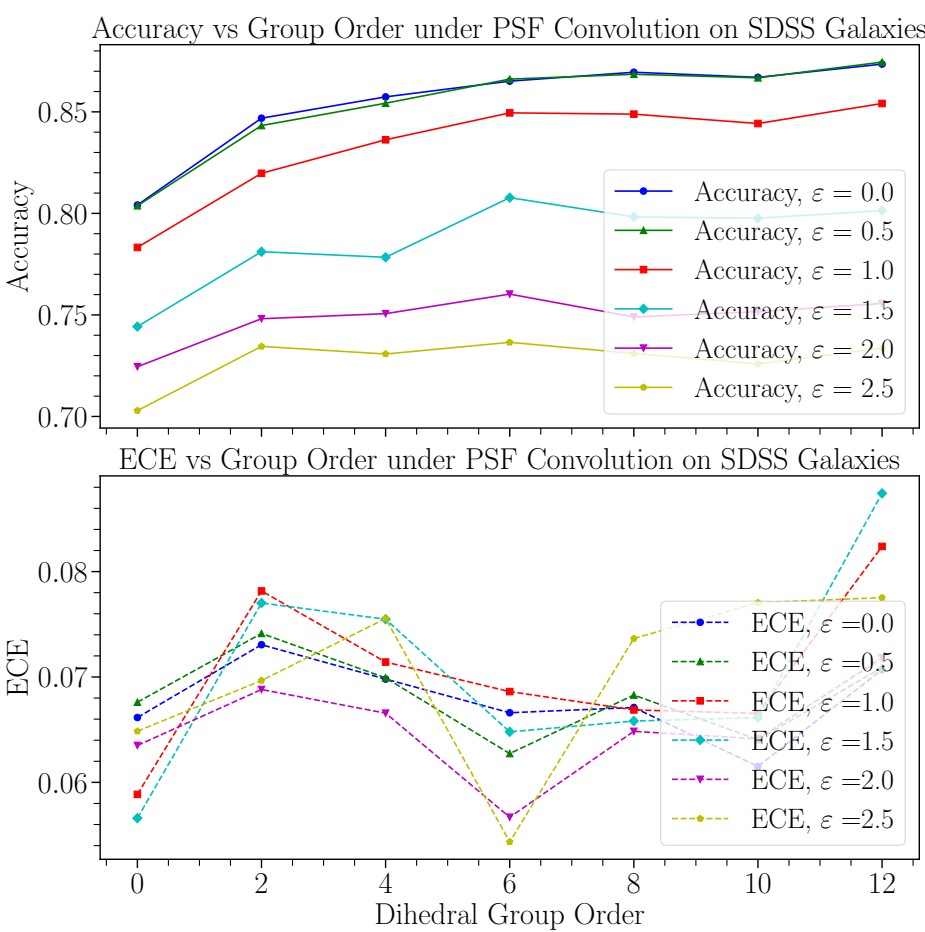

Figure 13: Accuracy and ECE vs Dihedral Group Order under PSF Convolution on SDSS Galaxies.

