# OpenReview forum: "On Uncertainty Calibration for Equivariant Functions"
_TMLR — Accepted by TMLR_

### Review · Reviewer_FFeh · 2025-11-20

**Summary Of Contributions:**

The authors introduce new theoretical and empirical contributions in the study of equivariant learning and uncertainty quantification. Particularly interesting is the authors' discussion of how achieving correct equivariance with a model may not necessarily reduce its aleatoric bleed (i.e., epistemic error disguised as aleatoric error). The authors demonstrate this finding through computational experiments and outline directions for future research in this domain, which may inspire additional considerations, e.g., for scientific applications of equivariant neural networks.

**Additional Comments:**

1. **Typos:** There is a typo in "equivariance" on line 3 of Section 2; one on Page 3 in "a reliable uncertainty estimates"; one in the first sentence of Section 4 ("for on"); one in "and to properly predicts" at the beginning of Section 4.1; one in "may very" in Remark 1; one in "corresponding to correspond" on Page 11; "illustrates the how" and "the which elements" on Page 12; "tjat" on Page 16; "on one eac" on Page 23; "is important it" on Page 24; caption "in QM9s" in Table 1; "biasses on Page 29; and "that reinforcement learning" on Page 30.
2. **Citations:** The normalizing flows citation list should include [1] on Page 3.
3. **Clarity:** (1) Can axes be labeled in Figure 2 to make the units of measure (e.g., `x`) clearer to readers? I found it difficult to tie some of the authors' descriptions of units of measurement to this figure when they were described in the main text. (2) How are the decoder heads typically supervised for parametrizing the Student's $t$ distribution in Section 6.1.2? What are the nuances (or tradeoffs) in deriving these ground-truth labels?

**References:**

[1] Rezende, D., & Mohamed, S. (2015, June). Variational inference with normalizing flows. In International conference on machine learning (pp. 1530-1538). PMLR.

**Audience:**

Yes

**Audience Explanation:**

Equivariance and uncertainty quantification are both areas of considerable interest in the machine learning community. Especially in the area of AI for Science, joint knowledge of these two disciplines may become more relevant over time.

**Broader Impact Concerns:**

I have no concerns regarding broader impacts.

**Claims And Evidence:**

Yes

**Claims Explanation:**

1. The authors introduce a generalized version of the Expected Normalized Calibration Error (GENCE) supporting vector-valued data.
2. The authors perform computational experiments illustrating trends such as test accuracy improvement as a function of equivariance adherence. Interestingly, Expected Calibration Error (ECE) doesn't always display similar kinds of trends.
3. The authors' experiments involving scientific datasets are well-warranted and help ground the authors' study in recent trends in AI for Science (e.g., investigating how and when to learn equivariance vs. building it into one's model architecture).

**Requested Changes:**

My main concern regards the presentation of the authors' theoretical and empirical findings. As a machine learning practitioner, the theory the authors discuss and introduce provides helpful context yet consumes a lot of space preceding the authors' computational experiments. I personally found it difficult to read through these initial sections (i.e., Sections 4-6), but this might be due in part to my lack of strong expertise in uncertainty quantification. Overall, I think it would be helpful to consider adding a short note (in these early sections) to readers who are mainly interested in the authors' empirical results and conclusions, so that they will know at which point in the main text they can safely skim or skip ahead to pages 20+ for the proceeding benchmarking studies. Or better yet, foreshadowing in Sections 4-6 some of the authors' empirical findings might better motivate these derivations for unfamiliar readers.

---

> ### Author Response · Authors · 2025-12-11
> **Presentation Changes**
>
> Thank you for your thoughtful review.
>
> ## Requested Changes
>
> > Overall, I think it would be helpful to consider adding a short note (in these early sections) to readers who are mainly interested in the authors' empirical results and conclusions, so that they will know at which point in the main text they can safely skim or skip ahead to pages 20+ for the proceeding benchmarking studies. Or better yet, foreshadowing in Sections 4-6 some of the authors' empirical findings might better motivate these derivations for unfamiliar readers.
>
> Per your suggestion, we have inserted text (given in blue) that helps guide the reader in finding the empirical results and hints at the results in the beginning of the classification and regression sections. For forward foreshadowing, we added a sentence in example 2 that references the latter experimental sections.
>
> ## Additional Comments 1-3
>
> We have addressed all the typographical errors and edits, included the appropriate citation, and updated the figure axes in Figure 2. The x-axis now reads $ | $ { $ x : f(x) = f(gx) $ } $/ |${$x$}$|$. In this setting, the group element $g$ is just the cyclic permutation of the z component.
>
> > How are the decoder heads typically supervised for parameterizing the Student's t distribution in Section 6.1.2?
>
> This is explained in Appendix E. Following Amini et al. 2020 [1], we take the negative log likelihood of the distribution in order to derive a surrogate loss as a function of the four parameter family. The main design choice was that we used a different decoder head for each parameter $(\alpha, \beta, \nu, \gamma)$. We found this to be more reliable than having a single decoder head estimate all four parameters simultaneously, however, the latter is a valid choice in principle. For example, Pandya et al. [2] uses single decoder heads to estimate uncertainty parameters simultaneously. Beyond this decision, our methodology follows the typical supervision strategy for estimating the parameters of a probability distribution.
>
> > What are the nuances (or tradeoffs) in deriving these ground-truth labels?
>
> We assume your question is asking where the true labels for $\alpha, \beta, \nu, \gamma$ come from. Only the labels for $y$ are provided. It is assumed that $y$ is drawn from a distribution in the evidential regression setup, and thus all four parameters are needed to fit this distribution. We are only able to compute the aleatoric bleed because we choose a setting where we know that all of the uncertainty should be epistemic. If we have misinterpreted your question, please let us know.
>
> [1] Amini et al. Deep Evidential Regression. NeurIPS 2020. https://proceedings.neurips.cc/paper_files/paper/2020/file/aab085461de182608ee9f607f3f7d18f-Paper.pdf
>
> [2] Pandya et al. IAEmu: Learning Galaxy Intrinsic Alignment Correlations. Open Journal of Astrophysics 2025. https://arxiv.org/abs/2504.05235

---

> > ### Comment · Reviewer_FFeh · 2025-12-12
> > **Response to author rebuttals**
> >
> > I'd like to thank the authors for their prompt responses. Overall, I believe they address my primary concerns and questions. The only remaining comment I have is that Figure 2's caption appears to have a typo: "the model is invariant $C_2$ reflections...". Further, I believe the authors were referring to Figure 3 when they mentioned x-axis relabeling.

---

> ### Author Response · Authors · 2025-12-12
> **Correcting Figure**
>
> Thank you for pointing this out! We amended the caption in Figure 2 to read
>
> > Dataset with pointwise incorrect invariance. The first two panels indicate that the model is invariant to the action of $C_2$, which performs reflections over the $x$-axis.
>
> Furthermore, have posted a revised pdf with a new Figure 2 that contains axis labels and a tick mark to indicate the x=1 point to clarify the set up for readers.

---

> > ### Comment · Reviewer_FFeh · 2025-12-12
> > **Response to figure correction**
> >
> > Thanks again to the authors for their quick response. The new Figure 2 looks better to me now.

---

### Review · Reviewer_AnLE · 2025-11-23

**Summary Of Contributions:**

This paper addresses an important gap in the geometric deep learning literature: the relationship between equivariance and model calibration. The authors generalizes a theoretical framework that connects symmetry constraints to uncertainty estimation baed on the theoretical framework from Wang et al. (2024). They derive lower and upper bounds for Expected Calibration Error (ECE) in classification and Expected Normalized Calibration Error (ENCE) in regression, specifically under conditions of symmetry mismatch (correct, incorrect, and extrinsic equivariance). Additionally, the paper introduces "Generalized ENCE" (GENCE) for vector-valued regression and a novel metric, "Aleatoric Bleed," to measure the conflation of epistemic and aleatoric uncertainties.

**Audience:**

Yes

**Audience Explanation:**

Yes, this paper is likely to be of significant interest to several sub-communities within the journal's audience:

Geometric Deep Learning Researchers: This work addresses a fundamental gap in the literature by being the first to provide a rigorous theory connecting equivariance to uncertainty estimation. Researchers focusing on the theoretical properties of equivariant networks will find the fiber-wise error analysis particularly novel.

Machine Learning Practitioners: The paper explicitly targets domains like robotics, molecular physics, and astrophysics—fields where both data sparsity (addressed by equivariance) and safety/reliability (addressed by calibration) are critical. The insights on why equivariant models might fail to be calibrated despite high accuracy are practically valuable for these applications.

Uncertainty Quantification Community: The introduction of the "Generalized ENCE" (GENCE) metric for vector-valued regression and the concept of "Aleatoric Bleed" offers new tools for researchers looking to evaluate uncertainty in high-dimensional output spaces.

**Claims And Evidence:**

Yes

**Claims Explanation:**

Yes, the claims are generally well-supported by a combination of rigorous theoretical derivations and diverse experimental validation.

Theoretical Evidence: The authors provide accurate mathematical proofs deriving lower and upper bounds for calibration error (ECE and ENCE) by analyzing fiber-wise errors. This theoretical framework convincingly establishes the link between symmetry mismatch (specifically incorrect and extrinsic equivariance) and miscalibration.

Empirical Evidence: The submission includes a strong mix of synthetic and real-world experiments. The "Swiss Roll" and "Vector Field" experiments clearly isolate the effects of symmetry mismatch in a controlled setting. The "Galaxy Zoo" experiment provides convincing evidence for the paper's more nuanced claim: that increasing equivariance (via group order) improves accuracy but does not strictly guarantee better calibration.

Caveats: While the evidence for the existence of these relationships is clear, the practical application of the theoretical bounds is somewhat limited by their dependence on difficult-to-compute quantities (e.g., Lipschitz constants). Additionally, the evidence regarding "Aleatoric Bleed" in real-world chemical property tasks (QM9) was less decisive than in synthetic settings, though the authors transparently acknowledge this limitation.

**Requested Changes:**

Minor Comments

* Page 1, Abstract: There is a subject-verb agreement error in the sentence "relationships ... has yet to be studied." It should be "have yet to be studied."
* Page 2, Section 1: "dependant" is misspelled; it should be "dependent."
* Page 2, Section 2: There is a typo "equivaraince" which should be "equivariance."
* Page 4, Section 3.1: Missing article in "Let G be group." It should read "Let G be a group."
* Page 16, Section 4.6: There is a typo "tjat" which should be "that."
* Page 28, Figure 8 Caption: "fater tail" should be "fatter tail."
* Page 30, Section 9: "indespensible" is misspelled; it should be "indispensable."

Structural Suggestions

* Placement of Evidential Regression (Section 3.5): This section defines the specific parameterization used for the aleatoric uncertainty experiments. However, this framework is utilized almost exclusively in Section 6. To improve the flow and keep the definitions close to their application, I suggest moving the content of Section 3.5 to the beginning of Section 6 (e.g., as Section 6.1 "Preliminaries"). This would streamline the general background section.
* Ordering of Regression Theorems (Section 5): Currently, the flow of theoretical results in Section 5 is interrupted by "Example 6" (Section 5.4), which is placed between the Upper Bound theorems (5.2, 5.3) and the Lower Bound theorem (5.5). It would be more cohesive to present the theoretical bounds contiguously. I recommend moving Example 6 to the end of the section (after the Lower Bound theorem) to serve as a comprehensive illustration of the preceding theory.

---

> ### Author Response · Authors · 2025-12-11
> **Structural and Minor Changes**
>
> We thank you for your accurate assessment of our work. We have revised the manuscript to address all the minor edits above.
>
> > Placement of Evidential Regression (Section 3.5)
>
> We agree that the background from section 3.5 is far removed from the actual content in section 6. In our initial draft, we attempted to remind the reader of the relevant notation from the background, referencing section 3.5 in section 6. In revision, we incorporated your suggestion and repeated more of  the preliminaries in section 3.5 into section 6. The requested changes are given in blue.
>
> > Ordering of Regression Theorems
>
> Thank you for this suggestion. In the revision, we have moved the example to the requested location.

---

> > ### Comment · Reviewer_AnLE · 2025-12-13
> > **Response to Structural and Minor Changes**
> >
> > I appreciate the authors' responses. The revised version addressed my comments.

---

### Review · Reviewer_eEC3 · 2025-12-01

**Summary Of Contributions:**

## Summary
The paper studies how group equivariance and symmetry mismatch affect uncertainty calibration. Building on a taxonomy of correct, incorrect, and extrinsic equivariance, the authors derive population-level bounds on expected calibration error (ECE) for classification and introduce a generalized expected normalized calibration error (GENCE), together with a squared variant, for regression. These results characterize how calibration can deteriorate when the symmetries enforced by the predictor do not match those of the data-generating process, extending earlier work that focused primarily on prediction error under symmetry mismatch.

Within an evidential regression framework, the paper formalizes aleatoric and epistemic uncertainty via the parameters of a Normal–Inverse-Gamma prior and proposes aleatoric bleed as a quantitative measure of the leakage of epistemic uncertainty into the aleatoric component under symmetry mismatch. The theoretical development is complemented by empirical studies on synthetic vector-field and classification tasks, as well as galaxy morphology and molecular property prediction, using equivariant, incorrectly equivariant, extrinsically equivariant, and non-equivariant architectures. Collectively, the results provide an initial systematic account of how symmetry assumptions influence both calibration and the decomposition of predictive uncertainty.

## Strengths
1.	The paper offers a contribution by bringing calibration and uncertainty decomposition into the symmetry-mismatch story. Its main strength is conceptual: it takes seriously the question of how group equivariance and, crucially, incorrect or extrinsic equivariance affect not just prediction error but the calibration of predictive uncertainty. The authors derive population-level bounds on classification ECE and introduce GENCE (and a squared variant) as a population-level analogue of ENCE for vector-valued regression.
2.	The paper formalizes aleatoric and epistemic uncertainty through the parameters of a Normal–Inverse-Gamma prior and proposes “aleatoric bleed” as a diagnostic of how epistemic uncertainty can be spuriously reallocated into the aleatoric channel under symmetry mismatch.
3.	The empirical studies are substantial, spanning from synthetic vector fields and classification to galaxy morphology and molecular property prediction. The experiments demonstrate that symmetry mismatch can substantially degrade calibration and distort the aleatoric/epistemic split even when accuracy appears comparable.

## Weakness
1.	The presentation of the paper is confusing and inconsistent in Section 3. I suggest the authors to carefully address the following clarity problems and make adjustments to improve the paper's clarity overall. Clarity problem #1: In Definition 2, the three modes of equivariance were given with respect to the hypothesis class h, while the connection between the group G and h was never given. I need to find the exact definition in the paper (Wang et el. [1]). It should be stated earlier that ``For h that is equivariant under G...''.
2.	Clarity problem #2: It would be better to explicitly state in Theorem 2 and 3 that the error function as $err_{reg}(h)$ to avoid confusion.
3.	Clarity problem #3: In Theorem 3, the orbit-wise density $\alpha(g, x)$ was never defined in the main text. The only place where $\alpha$ is actually defined is Appendix A: Iterated Integration.
4. Clarity problem #4: The paper explains the aleatoric uncertainty and epistemic uncertainty in the related work section (Page 3), while this should at least appear in the main text such as Sec 3.5 and Sec 6.
5. The main weakness is that the paper contributes much more conceptually, while providing limited theoretical insights. The theory is formulated entirely at the population level under fairly idealized measure-theoretic assumptions, with only cursory discussion of discrete data and finite-sample estimators, despite calibration metrics being notoriously delicate in practice.
6. GENCE is an appealing formal object, but the paper does not seriously engage with how one should estimate it from finite data, nor does it clearly situate GENCE with respect to existing regression-calibration and coverage-based approaches, so its added value beyond a more elegant restatement of ENCE remains somewhat underdeveloped.
7. The evidential treatment of aleatoric and epistemic uncertainty is precise and useful for analysis, yet the fundamental identifiability issue means that “aleatoric bleed” is unavoidably defined relative to a specific parametrization; the paper acknowledges this but still tends to treat the resulting metric as if it were close to canonical, and it provides limited comparison to alternative uncertainty-decomposition frameworks (e.g. Bayesian ensembles or mutual-information-based diagnostics).

Wang, Dian, et al. "A general theory of correct, incorrect, and extrinsic equivariance." Advances in Neural Information Processing Systems 36 (2023): 40006-40029.

**Audience:**

Yes

**Audience Explanation:**

The paper provides novel insights into bringing calibration and uncertainty decomposition into the symmetry-mismatch story. It formalizes the problem how group equivariance, incorrect or extrinsic equivariance affect the calibration of predictive uncertainty---which remains largely unstudied and is relevant to the audience in the field.

**Broader Impact Concerns:**

Not applicable.

**Claims And Evidence:**

Yes

**Claims Explanation:**

1.	The paper makes mainly conceptual contributions and the theoretical results are sound as far as I checked.
2.	The numerical experiments are thorough and substantial.

**Requested Changes:**

1.	Please state explicitly equivariance of h in Definition 2.
2.	Please explicitly state Theorem 2 and 3 that the error function are regression errors, using the notation $err_{reg}(h)$.
3.	Please move the definition of $\alpha(g, x)$ before Theorem 3.
4.	Explicitly (formally or informally) define the aleatoric uncertainty and epistemic uncertainty in Sec 3.5 and Sec 6.

---

> ### Author Response · Authors · 2025-12-11
> **Clarity Improvements and Addressing the Weaknesses**
>
> Thank you for your thorough and constructive review of our paper. We address the clarity issues and requested changes one at a time below. All requested changes are indicated in blue in our revised document.
>
> > Clarity problem #1: In Definition 2, the three modes of equivariance were given with respect to the hypothesis class h, while the connection between the group G and h was never given. I need to find the exact definition in the paper (Wang et el. [1]). It should be stated earlier that ``For h that is equivariant under G...''.
>
> Thank you for pointing this out. We have now stated explicitly in definition 2 that h is G-equivariant.
>
> > Clarity problem #2: It would be better to explicitly state in Theorem 2 and 3 that the error function as err reg to avoid confusion.
>
> We agree that this aids in the presentation of the theorems and have incorporated this comment.
>
> > Clarity problem #3: In Theorem 3, the orbit-wise density $\alpha(g,x)$ was never defined in the main text. The only place where is actually defined is Appendix A: Iterated Integration.
>
> We thank the reviewer for pointing this out. We have now defined $\alpha$ in text before theorem 3.
>
> > Clarity problem #4: The paper explains the aleatoric uncertainty and epistemic uncertainty in the related work section (Page 3), while this should at least appear in the main text such as Sec 3.5 and Sec 6.
>
> We thank the reviewer for this suggestion. In revision, we have included two definitions of aleatoric and epistemic uncertainty that are independent of any specific parameterization of the uncertainties, sourced from [1]. We include these definitions in sec 3.5, and repeat them in sec 6. for convenience.
>
> We also respond to the main weaknesses below.
>
> > The main weakness is that the paper contributes much more conceptually, while providing limited theoretical insights. The theory is formulated entirely at the population level under fairly idealized measure-theoretic assumptions, with only cursory discussion of discrete data and finite-sample estimators, despite calibration metrics being notoriously delicate in practice. GENCE is an appealing formal object, but the paper does not seriously engage with how one should estimate it from finite data, nor does it clearly situate GENCE with respect to existing regression-calibration and coverage-based approaches, so its added value beyond a more elegant restatement of ENCE remains somewhat underdeveloped.
>
> We agree that we made idealized assumptions as is common in ML theory in order to be able to derive prescriptive theorems. While we agree that our discussion of GENCE would benefit from advice on how to practically estimate it, we note that estimating ENCE-like quantities is already difficult in one dimension [2] and that push-forward densities are even harder to estimate in higher dimensions. Given these technical challenges, we feel that estimating GENCE is more appropriate for a follow up work focused on a more empirical investigation of the relationship between equivariance and uncertainty calibration than what is provided here. We also highlight that our latter experiments on aleatoric bleed empirically assess the effect of equivariance on calibration through a different lens.
>
> We agree with the reviewer that our paper would benefit from a discussion of other regression-calibration and coverage metrics. In our revised manuscript, we include a brief discussion of an alternative formulation. In this work, we use GENCE because it allows us to continue our discussion of calibration error bounds by noting that the calibration function is a fiber-wise error, but future work may consider alternative approaches. In addition, we highlight that our GENCE metric is more than a restatement of ENCE. It’s defined independently of any binning approximations and works for multivariate Gaussians, enabling new theoretical analysis. Subsequent works focused on empirically estimating the calibration statistics may also use our metric for multidimensional regression problems.

---

> > ### Author Response · Authors · 2025-12-11
> > **Addressing Clarity Concerns and Weaknesses Continuted**
> >
> > > The evidential treatment of aleatoric and epistemic uncertainty is precise and useful for analysis, yet the fundamental identifiability issue means that “aleatoric bleed” is unavoidably defined relative to a specific parametrization; the paper acknowledges this but still tends to treat the resulting metric as if it were close to canonical, and it provides limited comparison to alternative uncertainty-decomposition frameworks (e.g. Bayesian ensembles or mutual-information-based diagnostics).
> >
> > We agree that a more comprehensive comparison between different uncertainty decomposition approaches would strengthen the paper. In this work, we chose evidential regression because we find it to be the most grounded theoretically in the case of a zero ground-truth aleatoric uncertainty compared to alternatives. In particular, we want our decomposition strategy to be generally robust in the setting with zero symmetry mismatch so that we can isolate the effect of aleatoric bleed to symmetry mismatch in our experiments.
> > We note that the suggested alternative of mutual information has been shown to violate several standard properties for uncertainty decomposition [3], including when epistemic uncertainty is maximal and how epistemic uncertainty behaves under mean preserving and variance preserving spreads of the data. While evidential regression methods have their own drawbacks [4], notably convergence issues that require careful attention when training, we find the theoretical drawbacks of other proposed methods to be more severe. In particular, we note the following observation in [3]: Discrepancies between growth truth scenario aleatoric uncertainties with zero epistemic uncertainty and the finite-sampling setting in which aleatoric uncertainty is derived from (and hence depends on) epistemic uncertainty leads to incoherences and calls the decomposition of uncertainties into question.
> >
> > This issue would undermine the conclusions drawn from our experiments, since we explicitly construct our experiments under the assumption of a ground-truth aleatoric uncertainty. Moreover, in the case of no symmetry mismatch, there should be very little epistemic uncertainty. The work of [3] highlights that the decomposition is poor even without any symmetry mismatch, and similarly shows experimentally that ensembling is also insufficient. In contrast, our adoption of evidential regression shows expected behavior when the symmetry constraint is correct. We also performed ablations on hyperparameters in order to be sure that our evidential regression models properly converged in order to counteract the potential training pitfalls laid out in [4].
> >
> > We agree with the reviewer that the paper lacks adequate contextualization and a clear framing of our modeling choices. In the revised manuscript, our added definitions of aleatoric and epistemic uncertainty and following discussion now clarify that evidential regression is one such decomposition of the two uncertainties among other techniques and parameterizations, and we have updated our text to reflect why we choose evidential regression. We also note that our text already defines aleatoric bleed absent of any parameterization, and then describes how one would compute aleatoric bleed in the specific case of evidential regression.
> >
> > [1] Eyke Hüllermeier and Willem Waegeman. Aleatoric and epistemic uncertainty in machine learning: An introduction to concepts and methods. Machine learning, 110(3):457–506, 2021. https://arxiv.org/pdf/1910.09457
> >
> > [2] Pascal Pernot. Properties of the ENCE and other MAD-based calibration metrics. Arxiv preprint. https://arxiv.org/abs/2305.11905
> >
> > [3] Wimmer et al. Quantifying Aleatoric and Epistemic Uncertainty in Machine Learning: Are Conditional Entropy and Mutual Information Appropriate Measures? Uncertainty in Artificial Intelligence 2023. https://openreview.net/pdf?id=OOfLGZQVSk
> >
> > [4] Juergens, Mira, et al. "Is Epistemic Uncertainty Faithfully Represented by Evidential Deep Learning Methods?." International Conference on Machine Learning. PMLR, 2024. https://arxiv.org/abs/2402.09056

---

> > > ### Comment · Reviewer_eEC3 · 2025-12-13
> > > **Replying to authors responses**
> > >
> > > I appreicate the authors detailed responses. It addresses my concerns, and I'm happy to see the revised version.

---

### Decision · Action_Editor_hgJo · 2025-12-28

**Recommendation:** Accept as is

**Audience:**

Yes

**Audience Explanation:**

The paper addresses a question that is of clear interest to the TMLR audience: how equivariance affects uncertainty calibration. This topic is relevant to researchers working on geometric deep learning, uncertainty estimation, and applications in scientific domains where reliability matters. The combination of theory and experiments makes the findings accessible and useful to readers with both theoretical and applied interests.

**Claims And Evidence:**

Yes

**Claims Explanation:**

The claims are supported by solid theoretical analysis and well-designed experiments. The paper provides clear theory and proofs linking equivariance and symmetry mismatch to calibration error, and the experiments on both synthetic and real datasets consistently reflect these theoretical findings. The reviewers agreed that the evidence is convincing, and the authors have addressed the earlier clarity issues in the revision.

---

> ### Author Response · Authors · 2026-01-28
> **Submitted CR**
>
> Dear Action Editor, thank you for a great review process! We have just submitted our camera ready version!